# TILT MATCHING FOR SCALABLE SAMPLING AND FINE-TUNING

## ABSTRACT

We propose a simple, scalable algorithm for using stochastic interpolants to perform sampling from unnormalized densities and for fine-tuning generative models. The approach, Tilt Matching, arises from a dynamical equation relating the velocity field for a flow matching method to the velocity field that would target the same distribution tilted by a reward. As such, the new velocity inherits the regularity of stochastic interpolant transport plans while also being the minimizer of an objective function with strictly lower variance than flow matching itself. The update to the velocity field that emerges from this simple regression problem can be interpreted as the sum of all joint cumulants of the stochastic interpolant and copies of the reward, and to first order is their covariance. We define two versions of the method, Explicit and Implicit Tilt Matching. The algorithms do not require any access to gradients of the reward or backpropagating through trajectories of the flow or diffusion. We empirically verify that the approach is efficient, unbiased, and highly scalable, providing state-of-the-art results on sampling under Lennard-Jones potentials and is competitive on fine-tuning Stable Diffusion, without requiring reward multipliers. It can also be straightforwardly applied to tilting few-step flow map models.

## 1 INTRODUCTION

Generative models built out of dynamical transport like flow and diffusion models are highly scalable tools that serve as building blocks for foundation models across industries (Rombach et al., 2022; Geffner et al., 2025; Watson et al., 2023; Brooks et al., 2024; Zeni et al., 2025). These models work by building a continuous time map connecting a base distribution to a target distribution, realized by solving a differential equation whose coefficients are outputs of neural networks.

There is now a vested interest in applying them in settings where there is not *a priori* an abundance of data to learn from to complete a task of interest. These include learning to sample under Boltzmann distributions appearing in molecular dynamics Noé et al. (2019); Herron et al. (2024); Plainer et al. (2025) and statistical physics Albergo et al. (2019); Gabrié et al. (2022); Kanwar et al. (2020); Nicoli et al. (2021), as well as fine-tuning an existing generative model so as to produce samples that align with user requests.

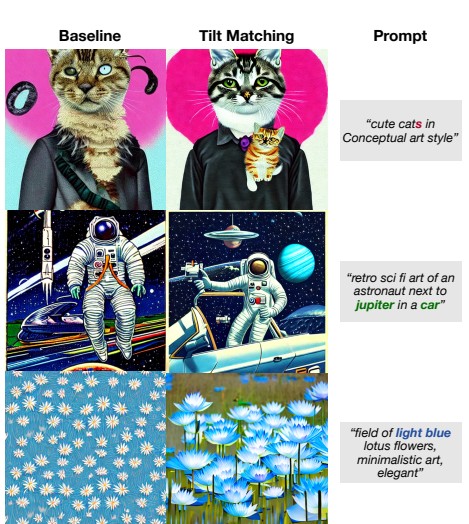

**Figure 1:** Example improvements to Stable Diffusion 1.5 using Tilt Matching and ImageReward.

Both of these problems can be framed as **_tilting_** some existing distribution toward a new target. For Boltzmann sampling, this means adapting the energy function defining the theory; for fine-tuning, this means adapting the base generative model to score highly against a reward $r(x)$. Our aims in this paper are precisely centered around this picture. Given access to samples from a distribution with density $\rho_1(x) : \mathbb{R}^d \to \mathbb{R}$, we want to learn to sample the tilted distribution $\rho_{1,a} \propto \rho_1 e^{ar(x)}$, where

$r(x)$ is a scalar function which defines the **tilt** and $a$ is an annealing parameter that characterizes the extent of the tilt. This initial distribution $\rho_1 = \rho_{1,a=0}$ could be given by an existing generative model, as in the case of fine-tuning, or it may be a reference distribution that is easy to sample with conventional techniques when performing sampling. We will ultimately be interested in $\rho_{t,a=1}$, i.e. the density fully tilted toward the reward.

While diffusions (Song et al., 2020; Ho et al., 2020) and flow-based models (Lipman et al., 2022; Albergo & Vanden-Eijnden, 2022; Liu et al., 2022) work well when data is available, regression objectives for the data-less contexts we focus on here are still not available, or come with caveats. In what follows, we briefly summarize the highly scalable generative models. Then, we will motivate a practical *modification* to these approaches so that they maintain many of their appealing optimization qualities while making them applicable to fine-tuning and sampling. To this end, we specify our **main contributions:**

- We derive an evolution equation for stochastic interpolant velocity fields under reward tilts that has a fundamental connection to the higher order moments between the interpolant and the reward.

- We show how the above fact allows us to construct ***Tilt Matching***, a family of simple iterative regression loss functions for the tilted velocity field that do not rely on backpropagating through generated trajectories, do not require spatial gradients of the reward, can avoid likelihood computations during training, and whose variances are strictly less than that of flow matching itself and can be further systematically improved with control variates.

- We instantiate two versions of the objective, Explicit Tilt Matching and Implicit Tilt Matching; the latter completely removes discretization errors from iteratively updating the tilt.

- We show how the method can be applied to both sampling distributions known up to normalizing constant and to fine-tuning existing generative models, where we achieve state-of-the-art performance on sampling Lennard-Jones potentials with diffusion based samplers, and can improve perceptual scores of Stable Diffusion 1.5 with a straightforward application of the algorithm.

## 2 RELATED WORK

**Neural Samplers**   Employing transport in Monte Carlo sampling algorithms has been an active research topic beginning with the work of Marzouk et al. (2016), and made parametric with neural networks in (Noé et al., 2019; Albergo et al., 2019), using coupling-based normalizing flows (Rezende & Mohamed, 2015; Dinh et al., 2017). Recent works have sought to perform this sampling with continuous time flow and diffusion models. These "neural samplers" take on various forms. Some approach the problem from an optimal control perspective (Zhang & Chen, 2022; Tzen & Raginsky, 2019; Havens et al., 2025) involving backpropagating through stochastic trajectories. Others interface with annealed importance sampling Neal (1993) and attempt to learn drift coefficients along a geometric annealing path either through trajectories (Vargas et al., 2024) or physics informed neural network (PINN) objectives (Máté & Fleuret, 2023; Tian et al., 2024; Albergo & Vanden-Eijnden, 2024; Holderrieth et al., 2025). The trajectory based losses can become unstable if the number of steps taken in solving the SDE is not sufficiently small, and while the PINN based losses avoid this, they can sometimes involve unstable or expensive terms based off of derivatives of neural networks in the loss function. Other works like (Vargas et al., 2023) also try to learn to sample along the time-dependent density of a diffusion process, but their formulation requires backpropagating through trajectories. Our proposed approach inherits the potential efficiency of coupling based flows because all of it can be defined with the any step flow map (Boffi et al., 2024; 2025; Sabour et al., 2025); does not rely on backpropagating through trajectories; does not require computation of likelihoods; and does not require enforcing a PINN loss with gradients of neural networks in it. A related method PTSD (Rissanen et al., 2025) iteratively trains a diffusion model along a temperature ladder, using approximate samples coming from a diffusion model using a finite-difference approximation of the score function. PTSD also employs reweighting and local parallel tempering refinement to reduce bias. These techniques are compatible with our framework, while the ITM objective avoids the finite-difference approximation error.

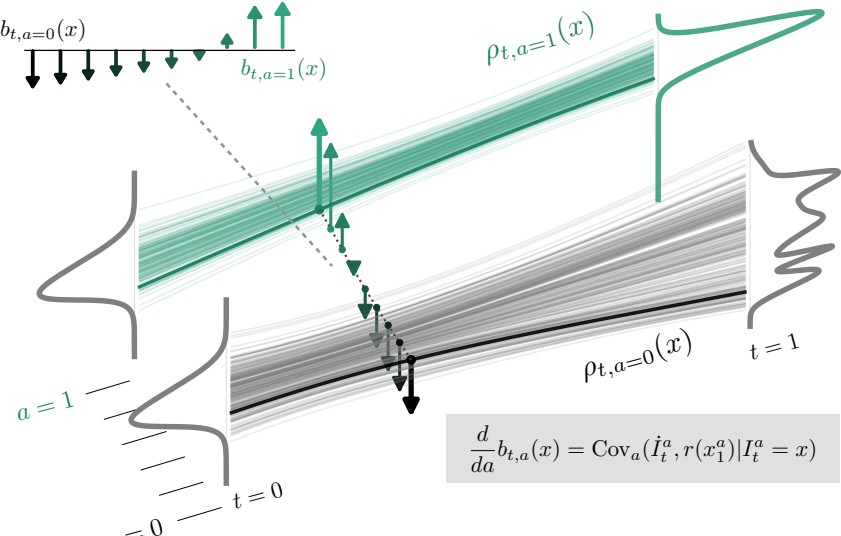

**Figure 2:** Schematic overview of the proposed method. When a stochastic interpolant is used to learn a generative model $b_{t,a=0}$ that samples $\rho_{t,a=0}$ and in particular $\rho_{1,a=0}$ (gray curve), then the evolution of that velocity field in $a$ in order to sample $\rho_{1,a>0} = \frac{1}{Z}\rho_{1,0}e^{ar(x)}$, where $r$ is a reward function, has closed form given by the covariance of the dynamics of the interpolant at $(t, a)$ and the reward. The velocity field, denoted as up or down arrows showing direction of motion in $x$, changes from negative to positive in the above toy example.

**Fine-tuning flows and diffusions**  Modifying the drift of the generative process is the predominant strategy for fine-tuning dynamical transport models. Existing work follows two high-level approaches: 1) reward-maximizing methods that directly optimize the quality of generated samples, such as D-Flow Ben-Hamu et al. (2024) and DRaFT Clark et al. (2024), and 2) distribution matching techniques that align the model with a reward-tilted distribution to prevent overfitting, seen in DEFT (Denker et al., 2024), adjoint matching (Domingo-Enrich et al., 2025), GFlowNet approaches (Zhang et al., 2024; Liu et al., 2025b), and approaches adapted from DPO Wallace et al. (2024). Nevertheless, these algorithms frequently suffer from major disadvantages, including the need to differentiate through trajectories Denker et al. (2024); Ben-Hamu et al. (2024); Clark et al. (2024) or the requirement of a differentiable reward function (Ben-Hamu et al., 2024; Clark et al., 2024; Zhang et al., 2024; Liu et al., 2025b; Domingo-Enrich et al., 2025), while some are only approximate Wallace et al. (2024). The proposed tilt matching method is free from these limitations.

## 2.1 DYNAMICAL TRANSPORT, STOCHASTIC INTERPOLANTS, AND FLOW MAPS

Many state-of-the-art generative models that aim to model a data distribution $\rho_1(x)$ learned from samples $\{x_1\}_{i=1}^N$ do so by means of dynamically mapping samples from a reference distribution $x_0 \sim \rho_0$. This mapping is defined by a drift coefficient in a flow Albergo & Vanden-Eijnden (2022); Lipman et al. (2022) or diffusion process Song et al. (2020); Ho et al. (2020), e.g. appearing in the ordinary differential equation (ODE)

$$\dot{x}_t = b_t(x_t), \qquad x_0 \sim \rho_0 \tag{1}$$

where $b_t : [0, 1] \times \mathbb{R}^d \to \mathbb{R}^d$ is a vector field that governs the transport such that the solution to equation 1 up to time $t$ produces a sample $x_t \sim \rho_t$. The PDF $\rho_t$ of this process satisfies the continuity equation

$$\partial_t \rho_t + \nabla \cdot (b_t \rho_t) = 0, \qquad \rho_{t=0} = \rho_0. \tag{2}$$

In generative modeling, our aims are to learn $b_t$ over neural networks such that the marginal law arising from (1) satisfy (2). A highly scalable, unifying perspective for dynamical transport models is that of stochastic interpolants Albergo & Vanden-Eijnden (2022); Albergo et al. (2023). A stochastic interpolant $I_t(x_0, x_1) : [0, 1] \times \mathbb{R}^d \times \mathbb{R}^d \to \mathbb{R}^d$ defined as

$$I_t := \alpha_t x_0 + \beta_t x_1 \qquad x_0, x_1 \sim \rho(x_0, x_1), \tag{3}$$

where $\alpha_t, \beta_t$ are functions of time satisfying $\alpha_0 = \beta_1 = 1$ and $\alpha_1 = \beta_0 = 0$, is a stochastic process such that $\mathrm{Law}(I_t) = \rho_t$. Importantly, the velocity field associated to this $\rho_t$ which solves equation 2 has a closed form which is given by $b_t(x) = \mathbb{E}[\dot{I}_t | I_t = x]$, where the expectation is taken over the coupling $(x_0, x_1) \sim \rho(x_0, x_1)$ conditional on $I_t = x$. Plugging this expression into a regression loss function to learn $b_t$ over neural networks gives, by tower property of the conditional expectation,

$$b_t = \arg\min_{\hat{b}_t} \int_0^1 \mathbb{E} \left| \hat{b}_t(I_t) - \dot{I}_t \right|^2 dt. \tag{4}$$

This procedure is the backbone of various large-scale generative models across various domains such as image and video generation Esser et al. (2024) and protein design Geffner et al. (2025). The main question to keep in mind going forward is: *how is the solution $b_t$ of one transport problem related to the solution of another?*

## 2.2 FINE TUNING AND SAMPLING AS TILTING

One might ask how this $b_t$ could be modified such that it solves the transport not for $\rho_1$, but rather the tilted distribution $\rho_{1,a}$ which defines our fine-tuning or sampling problem. That is, how are the velocity fields $b_{t,a=0}$ and $b_{t,a>0}$ related, and is there a learning paradigm that would allow us to estimate $b_{t,a}$ when initially given access only to the ground truth velocity field $b_{t,0}$ for the original generative model? This would allow us to ultimately evolve $b_{t,a}$ all the way to $b_{t,1}$, which would be the velocity field that can be used to directly sample the tilted distribution. We now introduce our method focused on this evolution, which we call **Tilt Matching Models (TMMs)**, a scalable procedure for adapting velocity fields under tilting.

## 3 DERIVING TILT MATCHING

To approach this question, consider modifying (3) so that it instead uses samples $x_1^a \sim \rho_{1,a}$

$$I_t^a := \alpha_t x_0 + \beta_t x_1^a \tag{5}$$

i.e. $\mathrm{Law}(I_t^a) = \rho_{t,a}$. Learning the velocity directly from this interpolant would be convenient, but we do not have samples a priori under $\rho_{t,a>0}$ to construct it, so this object is not immediately useful. However, it is possible to define $b_{t,a}$ in terms of the original interpolant, which we do have access to, combined with weights via:

$$b_{t,a}(x) = \frac{\mathbb{E}[\dot{I}_t^0 e^{ar(x_1)} | I_t^0 = x]}{\mathbb{E}[e^{ar(x_1)} | I_t^0 = x]}. \tag{6}$$

This relation is proven in the appendix, and it also straightforwardly holds for a shift of arbitrary size $h$ from $a$ to $a + h$:

$$b_{t,a+h}(x) = \frac{\mathbb{E}[\dot{I}_t^a e^{hr(x_1^a)} | I_t^a = x]}{\mathbb{E}[e^{hr(x_1^a)} | I_t^a = x]}. \tag{7}$$

If $h$ is large, then the variance of this expression may make any computational realizations of it impractical. Instead, by taking the derivative of (7) with respect to $a$, we can ask how $b_{t,a}$ should evolve to anneal it toward our target velocity field. The following proposition shows that the *evolution* of the velocity field $b_{t,a}(x)$ associated to equation 5 with respect to $a$ has a closed form defined solely in terms of known or learnable quantities:

**Proposition 1.** *(Covariance ODE.) Let $I_t^a = \alpha_t x_0 + \beta_t x_1^a$ be the interpolant constructed from samples $x_1^a \sim \rho_{1,a}(x)$. Then the augmented drift $b_{t,a}(x)$ satisfies*

$$\frac{\partial b_{t,a}(x)}{\partial a} = \mathbb{E}[\dot{I}_t^a r(x_1^a) | I_t^a = x] - b_{t,a}(x) \, \mathbb{E}[r(x_1^a) | I_t^a = x], \qquad b_{t,a=0}(x) = b_t(x) \tag{8}$$

*where the expectation is taken over the law of $I_t^a$ conditional on $I_t^a = x$. The right-hand side of this equation is the conditional covariance $\mathrm{Cov}_a(\dot{I}_t^a, r(x_1^a) \mid I_t^a = x)$.*

Proposition 1 is proven in Appendix A. The above relation can be interpreted as a dynamical formulation of the Esscher transform (Esscher, 1932) arising from (7). The Esscher transform characterizes how expectations evolve under exponential tiltings. Here, applied to stochastic interpolant velocity fields, tilting by $e^{hr(x)}$ induces a flow on $b_{t,a}$ whose infinitesimal generator is the conditional covariance between the interpolant and the reward. Importantly, this evolution of $b_{t,a}$ with respect to $a$ only depends on the current $b_{t,a}(x)$, the modified interpolant equation 5, and the reward.

## 3.1 Explicit Tilt Matching

**First order expansion.** Because we can use the current $b_{t,a}(x)$ (or its flow map equivalent) to produce samples $x_1^a$, this suggests that the corrections to $b_{t,a=0}$ that need to be learned to sample the true tilted density can be learned in an iterative fashion by discretizing equation 8. That is, for $0 < h \ll 1$, we can write an **explicit Euler discretization** as

$$b_{t,a+h}(x) = b_{t,a}(x) + h \frac{\partial b_{t,a}(x)}{\partial a} + \mathcal{O}(h^2) \tag{9}$$

$$= b_{t,a}(x) + h\left(\mathbb{E}\left[\dot{I}_t^a \, r(x_1^a) \,\middle|\, I_t^a = x\right] - b_{t,a}(x) \, \mathbb{E}[r(x_1^a) \,|\, I_t^a = x]\right) + \mathcal{O}(h^2). \tag{10}$$

This perspective highlights TM as an iterative, covariance-guided procedure: starting from $b_{t,0}$, one can generate successive updates $b_{t,h}, b_{t,2h}, \ldots, b_{t,1}$ that gradually transform the velocity field toward the fully tilted distribution. As $h \to 0$, this discretization recovers the continuous evolution in equation 8, ensuring convergence to the desired $b_{t,1}$. To formalize this, we introduce the **residual operator**:

$$T_{t,a,h}^{\text{ETM}} := b_{t,a}(I_t^a) + \underbrace{h\left(\dot{I}_t^a \, r(x_1^a) - b_{t,a}(I_t^a) \, r(x_1^a)\right)}_{\text{residual}}. \tag{11}$$

The following proposition shows that $b_{t,a+h}$ can be efficiently regressed and is first-order accurate using what we call **Explicit Tilt Matching** (ETM):

> **Proposition 2.** *(Explicit Tilt Matching.) Assume $a \mapsto b_{t,a}(x)$ is $C^1$ with $\partial_a b_{t,a}$ given by (8), and let $h > 0$. Then, the unique minimizer of the regression objective*
>
> $$\mathcal{L}_{a \to a+h}^{\text{ETM}}(\hat{b}) := \int_0^1 \mathbb{E} \left\| \hat{b}_t(I_t^a) - T_{t,a,h} \right\|^2 dt. \tag{12}$$
>
> *is given by*
>
> $$\hat{b}_{t,a+h}(x) = \mathbb{E}[T_{t,a,h} \,|\, I_t^a = x]. \tag{13}$$
>
> *As such, training $\hat{b}_{t,a+h}$ to optimality on (12) produces a first-order accurate Euler update of the tilted velocity. Iterating for $a_k = kh$ with samples $x_1^{a_k}$ drawn using the current model defines a consistent scheme that converges to $b_{t,1}$ as $h \to 0$ under the above regularity.*

This procedure is appealing because it gives a velocity field $b_{t,a=1}$ with favorable regularity conditions since the ultimate transport from $\rho_{t=0,a=1}$ to $\rho_{t=1,a=1}$ follows the interpolant path. This should make $b_{t,a=1}$ well-posed to be estimated with neural networks, as the transport for such paths starting from the Gaussian is geometrically smooth and does not exhibit any teleportation. At the same time, there are two main approximation errors to account for, which we outline next.

**Approximation error due to incomplete minimization of the objective.** A first source of error arises if the regression problem in (12) is not minimized exactly at each iteration. This means the learned drift $\hat{b}_{t,a}$ may deviate from the ideal $b_{t,a}$, with errors compounding over successive updates. To mitigate this, we can introduce importance weights during training, which corrects for residual mismatch between the model distribution and the tilted target. This strategy effectively debiases the procedure and prevents incomplete optimization from undermining convergence.

**Discretization error in $a$.** A second source of error comes from discretizing the Covariance ODE in the annealing parameter $a$. Since (8) defines a continuous evolution, replacing it with discrete

steps introduces a bias. In practice, this issue is negligible: we typically choose $h$ to be very small, so the resulting discretization bias can be almost completely eliminated. Moreover, the step size can be adapted dynamically using diagnostic quantities such as effective sample size (ESS) or changes in the estimated drift, which ensures the updates remain close to the continuous trajectory. However, computing diagnostics like the ESS can be computationally expensive, as it often requires calculating the divergence of the learned vector field. This motivates an alternative approach that eliminates discretization error by construction, rather than managing it with this adaptive scheme.

### 3.2 IMPLICIT TILT MATCHING

**Higher order expansions.** The explicit scheme defined by (12) arises by discretizing the evolution of $b_{t,a}$ given in (8) with a forward Euler step. While convenient, such updates inherit a discretization bias, which, even if small, might compound over successive steps in $a$. A natural extension is to consider taking higher-order Taylor expansions. Extending (9) to all orders gives

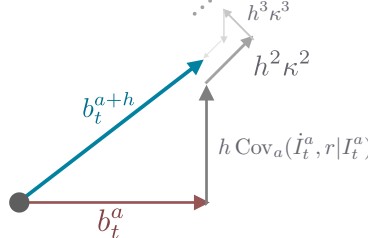

**Figure 3:** Pictorial additivity of higher order corrections to $b_{t,a+h}$. First order is the covariance, while higher order terms are cumulants $\kappa^n$.

$$b_{t,a+h}(x) = b_{t,a}(x) + \sum_{n>0} \frac{h^n}{n!} \frac{\partial^n}{\partial a^n}\big[b_{t,a}(x)\big]. \quad (14)$$

This statement on its own is contentless, but the following proposition shows that each term in the expansion has rich meaning:

**Proposition 3.** *(Tilt expansion.) For $b_{t,a} = \mathbb{E}[\dot{I}_t^a | I_t^a = x]$, the $n^{th}$ term $\frac{\partial^n}{\partial a^n}\big[b_{t,a}(x)\big]$ in the expansion in (14) is the $(n+1)^{th}$ order joint cumulant of the interpolant and $n$ instances of the reward, $\kappa^n(\dot{I}_t^a, r(x_1^a), \ldots, r(x_1^a))$.*

This result is proven in Appendix A and relies on a relation between the Esscher representation of $b_{t,a+h}$ and the moment generating function of the interpolant density. Since the cumulants involve higher order moments of the reward, a Monte Carlo training objective that attempts to match these term by term would be computationally infeasible.

**Expanding to all orders.** In order to fully eliminate the discretization error, we should consider all higher-order cumulants. Surprisingly, this expression is tractable as we show in the following proposition. If we define the **implicit residual operator** as

$$T_{t,a,h}^{\text{ITM}} := b_{t,a}(I_t^a) + \underbrace{\big(e^{hr(x_1^a)} - 1\big)\big(\dot{I}_t^a - b_{t,a+h}(x)\big)}_{\text{residual}}, \quad (15)$$

then we can learn the infinite cumulant expansion by directly matching against it:

**Proposition 4.** *(Implicit Tilt Matching.) Let $b_{t,a+h}$ be defined to all orders as in (14). Then*

$$\sum_{n>0} \frac{h^n}{n!} \frac{\partial^n}{\partial a^n}\big[b_{t,a}(x)\big] = \mathbb{E}[(e^{hr(x_1^a)} - 1)(\dot{I}_t^a - b_{t,a+h}(x)) | I_t^a = x] \quad (16)$$

*and $b_{t,a+h}$ is the minimizer of*

$$\mathcal{L}_{a \to a+h}^{\text{ITM}}(\hat{b}) := \int_0^1 \mathbb{E}\big\|\hat{b}_t(I_t^a) - T_{t,a,h}^{\text{ITM}}\big\|^2 dt, \quad (17)$$

*for any $h$, where expectation is taken over $(x_0, x_1^a) \sim \rho(x_0, x_1^a)$ conditional on $I_t^a = x$.*

This result *removes the discretization error* inherent to ETM and shows that all orders of the correction to the interpolant velocity field are directly learnable. Enforcing (16) is equivalent to enforcing that the residual update to $b_{t,a+h}$ is exact ***to all orders***. We call this condition **implicit** tilt matching

because the residual term that we add to $b_{t,a}$ depends on $b_{t,a+h}$ itself, leading to this fixed-point method.

The expression on the right-hand side of (16) may seem opaque, but it can be motivated with a simple derivation. Starting from the expression for the velocity $b_{t,a+h}$ in (7), we can multiply by the conditional expectation of the weight $\mathbb{E}[e^{hr(x)}|I_t^a = x]$ and rearrange terms to obtain the optimality condition for (17) in terms of $b_{t,a+h}$

$$\mathbb{E}\Big[e^{hr(x_1^a)}\big(b_{t,a+h}(x) - \dot{I}_t^a\big)\,\Big|\,I_t^a = x\Big] = 0. \tag{18}$$

Notice that if we take $h$ to be small, replace $e^{hr(x_1^a)} \approx 1 + hr(x_1^a)$ and replace one of the $b_{t,a+h}$ with $b_{t,a}$, then we recover the optimality condition of ETM. Thus we can view Implicit Tilt Matching as a generalization of the discretized procedure in (9) since its linearization recovers the ETM covariance update as specified by (9).

**Variance reduction via control variates.** We can further introduce a control variate $c(x) : \mathbb{R}^d \to \mathbb{R}$ into (18) to obtain a generalized optimality condition

$$\mathbb{E}\Big[c(x)\big(b_{t,a+h}(x) - b_{t,a}(x)\big) + \big(e^{hr(x_1^a)} - c(x)\big)\big(b_{t,a+h}(x) - \dot{I}_t^a\big)\,\Big|\,I_t^a = x\Big] = 0, \tag{19}$$

where we used the fact that $\mathbb{E}[c(x)\dot{I}_t^a|I_t^a = x] = \mathbb{E}[c(x)b_{t,a}(x)|I_t^a = x]$. The identity (19) holds for any choice of $c(x)$ and therefore suggests a family of valid implicit objectives we could use to find $b_{t,a+h}$. If we enforce (19) via a regression loss, the c-ITM objective would take the general form of

$$\mathcal{L}_{a\to a+h}^{\text{c-ITM}}(\hat{b}) = \int_0^1 \mathbb{E}\Big[e^{-hr(x_1^a)}\big\|c(I_t^a)\big(\hat{b}_t(I_t^a) - b_{t,a}(I_t^a)\big) + \big(e^{hr(x_1^a)} - c(I_t^a)\big)\big(\hat{b}_t(I_t^a) - \dot{I}_t^a\big)\big\|^2\Big]\,dt. \tag{20}$$

Alternatively, we can ensure (19) by finding the fixed point of the following stopgrad objective

$$\mathcal{L}_{a\to a+h}^{\text{c-sg-ITM}}(\hat{b}) = \int_0^1 \mathbb{E}\big\|c(I_t^a)\big(\hat{b}_t(I_t^a) - b_{t,a}(I_t^a)\big) + \big(e^{hr(x_1^a)} - c(I_t^a)\big)\big(\text{stopgrad}(\hat{b}_t(I_t^a)) - \dot{I}_t^a\big)\big\|^2\,dt. \tag{21}$$

Notice that the gradient of the latter objective is a $c(x)$ scaling of the former's gradients. The role of $c(x)$ is to control the variance of the Monte Carlo estimator of the loss function. Notice that the choice $c(x) = 1$ recovers (17) exactly. Moreover, this choice has the convenient property that for $h \ll 1$, it is close to the optimal control variate since $c(x) = 1$ clearly minimizes the variance conditional on $I_t^a = x$ when $h = 0$. (See the proof of Proposition 5.) More generally, one can optimize $c(x)$ to minimize the variance, yielding adaptive control variates that further stabilize training.

In practice when $h$ is very small, these higher order cumulants are likely negligible, and this process is still driven by the covariance. Nonetheless, it is now robust to any discretization errors, which we will explore experimentally later.

### 3.3 REWEIGHTING FLOW MATCHING VERSUS TILT MATCHING

In principle, the tilted drift $b_{t,a+h}$ could be obtained by applying flow matching directly to the interpolant $I_t^{a+h}$ with samples $x_1^{a+h} \sim \rho_{1,a+h}$:

$$b_{t,a+h} = \arg\min_{\hat{b}_t} \int_0^1 \mathbb{E}\Big[\big\|\hat{b}_t(I_t^{a+h}) - \dot{I}_t^{a+h}\big\|^2\Big]\,dt. \tag{22}$$

Since we do not have samples from $\rho_{t,a+h}$, the expectation in (22) must be expressed in terms of $\rho_{t,a}$, from which we do have samples. Introducing importance weights leads to what we call the *reweighted flow matching* (WFM) objective:

$$L_{a\to a+h}^{\text{WFM}}(\hat{b}) = \int_0^1 \mathbb{E}\Big[e^{hr(x_1^a)}\big\|\hat{b}_t(I_t^a) - \dot{I}_t^a\big\|^2\Big]\,dt. \tag{23}$$

Notice that this is precisely the c-ITM loss with $c(x) = 0$. Therefore WFM is an instantiation of c-ITM. As such, it has the same expected loss as any c-ITM variant. What differs between

the different algorithms is how the Monte Carlo estimates of the loss are taken: WFM regresses directly on $\dot{I}_t^a$, whereas ITM substitutes the dynamics of stochastic interpolant $\dot{I}_t^a$ with its conditional expectation $b_{t,a}(I_t^a)$. As such, ITM enjoys strictly lower variance than the WFM objective, at least for sufficiently small $h$:

> **Proposition 5.** *(Variance control).* *Let $\mathcal{L}_{a \to a+h}^{\mathrm{WFM}}$ and $\mathcal{L}_{a \to a+h}^{\mathrm{ITM}}$ be the regression losses in (23) and (12). For sufficiently small $h$, the gradient estimator of WFM has variance at least as large as that of ACM:*
>
> $$\mathrm{Var}\big[\nabla\mathcal{L}_{a \to a+h}^{\mathrm{WFM}}\big] \;\geq\; \mathrm{Var}\big[\nabla\mathcal{L}_{a \to a+h}^{\mathrm{ITM}}\big]. \tag{24}$$

This result formalizes that ITM enjoys a variance advantage over WFM because it centers updates on the conditional mean $b_{t,a}(I_t^a)$ rather than the noisy sample $\dot{I}_t^a$. We find this bound to have a meaningful implication in numerical experiments.

## 4 Numerical Experiments

An algorithm detailing the numerical implementation of ETM or ITM are given in Appendix C. In what follows, we test the proposed algorithms on both sampling Lennard-Jones (LJ) potentials (of 13 and 55 particles) and fine-tuning Stable Diffusion v1.5. For both setups, we build on existing code bases, e.g. from (Akhound-Sadegh et al., 2025) for the LJ experiments and (Domingo-Enrich et al., 2025; Blessing et al., 2025) for the fine-tuning. All network architectures are the same unless otherwise stated.

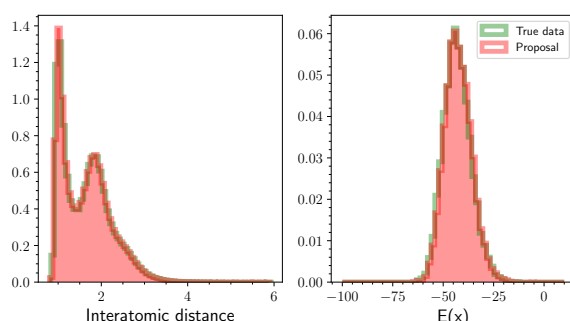

**Figure 4:** Comparison of LJ13 results using explicit tilt matching vs ground truth molecular dynamics data. **Left:** Histogram of interatomic distances amongst particles in the system. **Right:** Histogram of the energy of 10000 samples of the system. The method shows strong alignment with both measures.

### 4.1 Sampling Lennard-Jones potentials

In the context of sampling, the goal is to draw samples from a target density $\rho_1$, which is typically the Boltzmann distribution for a given potential energy function $E_1(x)$, such that $\rho_1(x) \propto \exp(-E_1(x))$. For TM, we begin with a simple prior density, $\rho_{1,a=0}$, which corresponds to an initial potential $E_0(x)$, and define an annealing path via linear interpolation:

$$E_a(x) = (1-a)E_0(x) + aE_1(x). \tag{25}$$

This defines a family of densities $\rho_{1,a}(x) \propto \exp(-E_a(x))$ for $a \in [0,1]$. This path is equivalent to the geometric annealing path described by the reward tilt formulation, where the reward is given by:

$$r(x) = E_0(x) - E_1(x) \tag{26}$$

A common choice for the prior $\rho_{1,a=0}$ is a Gaussian distribution. For molecular systems, a more effective strategy is to define the prior as a high-temperature analogue of the target by setting the initial potential as $E_0(x) = E_1(x)/T_0$, where $T_0 \gg 1$ is a high temperature. The resulting prior, $\rho_{1,a=0} \propto \exp(-E_1(x)/T_0)$, has a smoothed energy landscape that facilitates more efficient MCMC sampling. We adopt this temperature annealing approach for our numerical experiments.

We measure the performance of Tilt Matching against other methods by computing the effective sample size (ESS), the 2-Wasserstein distances on the energy and interatomic distance between the ground truth and our model outputs. We use the code from (Akhound-Sadegh et al., 2025) to perform the calculation. Note that their code attempts to replicate the Dist $\mathcal{W}_2$ as it appears in (Havens et al., 2025), which is also where the results for DDS and PIS come from, but that code is not available for exact reproduction.

We highlight a key advantage of ITM is its computational efficiency, as it performs well without requiring an adaptive annealing schedule. This contrasts with our implementation of ETM, where its

**Table 1:** Performance comparison on LJ-13 and LJ-55 using the effective sample size, 1D Energy histogram 2-Wasserstein and Distance 2−Wasserstein metrics. Missing −− entries indicate the metric is not applicable to that method or not available. We omit the ESS comparison for LJ-55 because it is too computationally intensive for us to compute, and other works do not provide a number to juxtapose with for similar reason.

| Method | LJ-13 | | | LJ-55 | |
| --- | --- | --- | --- | --- | --- |
| | ESS $\uparrow$ | $E(\cdot)\ \mathcal{W}_2 \downarrow$ | Dist $\mathcal{W}_2 \downarrow$ | $E(\cdot)\ \mathcal{W}_2 \downarrow$ | Dist $\mathcal{W}_2 \downarrow$ |
| DDS (Vargas et al., 2023) | 0.101 | 24.61 | 1.99 | 173.09 | 4.60 |
| PIS (Zhang & Chen, 2022) | 0.004 | 1.93 | 18.02 | 228.70 | 4.79 |
| iDEM (Akhound-Sadegh et al., 2024) | 0.231 | 1.352 | 0.127 | 93.53 | 4.69 |
| Adjoint Sampling (Havens et al., 2025) | −− | 2.40 | 1.67 | 58.04 | 4.50 |
| ASBS (Liu et al., 2025a) | −− | 1.28 | 1.59 | **27.69** | 4.00 |
| PITA (Akhound-Sadegh et al., 2025) | −− | 2.26 | 0.040 | – | – |
| ETM (Ours) | **0.740** | **0.270** | **0.012** | – | – |
| ITM (Ours) | **0.507** | 0.879 | **0.014** | 29.52 | **0.054** |

strong performance relies on an adaptive schedule guided by the ESS. The ESS calculation, however, requires computing the divergence of the learned vector field, which is a computationally intensive step. The overhead from this calculation made applying our adaptive ETM to larger systems such as LJ-55 impractical, highlighting that ITM is a more scalable and efficient algorithm.

## 4.2 Fine-tuning Stable Diffusion 1.5

To validate our proposed method, we finetune Stable Diffusion 1.5 (Rombach et al., 2022) using the ImageReward score (Xu et al., 2023) as the objective. Our implementation builds upon the codebase and parameters established in (Domingo-Enrich et al., 2025; Blessing et al., 2025). As our method operates within the stochastic interpolant framework (Albergo et al., 2023), we adopt the necessary transformations to recast the underlying denoising diffusion model, following the procedure detailed in the Appendix of (Domingo-Enrich et al., 2025).

To ensure a comprehensive evaluation and mitigate concerns of overfitting to a single reward metric, we additionally assess performance across three distinct axes: (1) text-to-image consistency, measured by CLIPScore (Hessel et al., 2021); (2) human aesthetic preference, quantified by HPSv2 (Wu et al., 2023); and (3) sample diversity, evaluated with DreamSim (Fu et al., 2023). We primar-

| Method | ImageReward ($\uparrow$) | ClipScore ($\uparrow$) | HPSv2 ($\uparrow$) | DreamSim ($\uparrow$) |
| --- | --- | --- | --- | --- |
| SD 1.5 (Base) | $0.1873 \pm 0.0762$ | $0.2746 \pm 0.0032$ | $0.2566 \pm 0.0030$ | $\mathbf{0.3849 \pm 0.0105}$ |
| AM ($\lambda = 1$) | $0.2170 \pm 0.0755$ | $0.2754 \pm 0.0032$ | $0.2576 \pm 0.0030$ | $0.3826 \pm 0.0104$ |
| **ETM** ($\lambda = 1$) | $0.3799 \pm 0.0744$ | $\mathbf{0.2801 \pm 0.0036}$ | $0.2655 \pm 0.0029$ | $0.3530 \pm 0.0118$ |
| **ITM** ($\lambda = 1$) | $\mathbf{0.4465 \pm 0.0709}$ | $0.2794 \pm 0.0036$ | $\mathbf{0.2659 \pm 0.0027}$ | $0.3383 \pm 0.0116$ |
| AM ($\lambda = 10^2$) | $0.7873 \pm 0.0689$ | $0.2792 \pm 0.0033$ | $0.2791 \pm 0.0028$ | $0.3363 \pm 0.0101$ |

**Table 2:** Finetuning results on Stable Diffusion 1.5. We compare our method against Adjoint Matching (Domingo-Enrich et al., 2025). We report on ClipScore (Hessel et al., 2021), HPSv2 (Wu et al., 2023), and DreamSim (Fu et al., 2023). For all metrics, higher values are better, as indicated by the up-arrow ($\uparrow$).

ily benchmark against adjoint matching (Domingo-Enrich et al., 2025), the current state-of-the-art for reward finetuning, which has demonstrated superior performance over prominent methods like DRaFT (Clark et al., 2024), DPO (Wallace et al., 2024), and ReFL (Xu et al., 2023). We emphasize that we finetune *only* on the ImageReward score, but measure performance on other scores as well.

Our results are summarized in Table 2, example images can be found in D. It is standard practice for adjoint matching to employ a reward multiplier, $\lambda$, which amplifies the reward signal to steer the learned distribution towards $\rho_1(x)e^{(\lambda \times r(x))}$. A key finding in our experiments is that our method achieves competitive performance against this strong baseline without a need for such a multiplier ($\lambda = 1$) or other hyperparameter tuning. This suggests that our approach provides a direct and stable mechanism for incorporating reward signals into the generation process, likely due to the fact that it

does not rely on spatial gradients of the reward for training or generation. Further gains could likely be made by hyperparameter sweeps.

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

## A  PROOFS

**Proposition 1.** *(Covariance ODE.) Let $I_t^a = \alpha_t x_0 + \beta_t x_1^a$ be the interpolant constructed from samples $x_1^a \sim \rho_{1,a}(x)$. Then the augmented drift $b_{t,a}(x)$ satisfies*

$$\frac{\partial b_{t,a}(x)}{\partial a} = \mathbb{E}[\dot{I}_t^a r(x_1^a)|I_t^a = x] - b_{t,a}(x)\,\mathbb{E}[r(x_1^a)|I_t^a = x], \qquad b_{t,a=0}(x) = b_t(x) \tag{8}$$

*where the expectation is taken over the law of $I_t^a$ conditional on $I_t^a = x$. The right-hand side of this equation is the conditional covariance $\mathrm{Cov}_a(\dot{I}_t^a, r(x_1^a) \,|\, I_t^a = x)$.*

*Proof.* To show equation 8, note that the pdf of $I_t^a$ is given explicitly as

$$\rho_{t,a}(x) = \mathbb{E}[\delta(x - I_t^0)e^{ar(x_1^0)}], \tag{27}$$

where $x_0 \sim \rho_0, x_1^0 \sim \rho_{1,0}$ and we ignore the normalizing constant as it will not be relevant for the proof. Taking the time derivative on both sides of this equation

$$-\nabla \cdot (b_{t,a}(x)\rho_{t,a}(x)) = -\nabla \cdot \mathbb{E}[\dot{I}_t^0 \delta(x - I_t^0)e^{ar(x_1^0)}] \tag{28}$$

where we assume $\rho_{t,a}(x) > 0$ else $b_{t,a}(x) = 0$. Isolating $b_{t,a}(x)$ gives

$$b_{t,a}(x) = \frac{\mathbb{E}[\dot{I}_t^0 \delta(x - I_t^0)e^{ar(x_1^0)}]}{\mathbb{E}[\delta(x - I_t^0)e^{ar(x_1^0)}]}. \tag{29}$$

Dividing the numerator and denominator by $\mathbb{E}[\delta(x - I_t^0)]$ proves equation 6. Taking its derivative with respect to a and using the quotient rule, we have

$$\frac{\partial}{\partial a}b_{t,a}(x) = \frac{\mathbb{E}[\dot{I}_t^0 \delta(x - I_t^0)e^{ar(x_1^0)}r(x_1^0)]}{\mathbb{E}[\delta(x - I_t^0)e^{ar(x_1^0)}]} - \frac{\mathbb{E}[\dot{I}_t^0 \delta(x - I_t^0)e^{ar(x_1^0)}]}{\mathbb{E}[\delta(x - I_t^0)e^{ar(x_1^0)}]}\frac{\mathbb{E}[\delta(x - I_t^0)e^{ar(x_1^0)}r(x_1^0)]}{\mathbb{E}[\delta(x - I_t^0)e^{ar(x_1^0)}]} \tag{30}$$

$$= \mathbb{E}[\dot{I}_t^a r(x_1^a)|I_t^a = x] - b_{t,a}(x)\,\mathbb{E}[r(x_1^a)|I_t^a = x], \tag{31}$$

which completes the proof. $\qquad\square$

**Proposition 2.** *(Explicit Tilt Matching.) Assume $a \mapsto b_{t,a}(x)$ is $C^1$ with $\partial_a b_{t,a}$ given by (8), and let $h > 0$. Then, the unique minimizer of the regression objective*

$$\mathcal{L}_{a \to a+h}^{\mathrm{ETM}}(\hat{b}) := \int_0^1 \mathbb{E}\left\|\hat{b}_t(I_t^a) - T_{t,a,h}\right\|^2 dt. \tag{12}$$

*is given by*

$$\hat{b}_{t,a+h}(x) = \mathbb{E}[T_{t,a,h} \,|\, I_t^a = x]. \tag{13}$$

*As such, training $\hat{b}_{t,a+h}$ to optimality on (12) produces a first-order accurate Euler update of the tilted velocity. Iterating for $a_k = kh$ with samples $x_1^{a_k}$ drawn using the current model defines a consistent scheme that converges to $b_{t,1}$ as $h \to 0$ under the above regularity.*

*Proof.* By the Hilbert $L^2$ projection theorem, among all functions of $I_t^a$, the optimizer is the conditional expectation $\mathbb{E}^a[T_{t,a,h} \,|\, I_t^a = x]$ where again $\mathbb{E}^a$ denotes expectation over the coupling $(x_0, x_1^a)$. Expanding $b_{t,a+h} = b_{t,a} + h\,\partial_a b_{t,a} + \mathcal{O}(h^2)$ and using (8) yields the expression above. $\quad\square$

**Proposition 3.** *(Tilt expansion.) For $b_{t,a} = \mathbb{E}[\dot{I}_t^a|I_t^a = x]$, the $n^{th}$ term $\frac{\partial^n}{\partial a^n}[b_{t,a}(x)]$ in the expansion in (14) is the $(n+1)^{th}$ order joint cumulant of the interpolant and $n$ instances of the reward, $\kappa^n(\dot{I}_t^a, r(x_1^a), \ldots, r(x_1^a))$.*

*Proof.* For a fixed $x$, define the joint conditional cumulant generating function of $r(x_1^a)$ and $\dot{I}_t^a$ as

$$M(\mu, \nu) = \log \mathbb{E}\left[e^{\mu r(x_1^a) + \langle \nu, \dot{I}_t^a \rangle}|I_t^a = x\right], \tag{32}$$

for $\mu \in \mathbb{R}$ and $\nu \in \mathbb{R}^d$. Its partial derivative with respect to $\nu$ evaluated at 0 is

$$\frac{\partial}{\partial \nu} M(\mu, 0) = \frac{\mathbb{E}[\dot{I}_t^a e^{\mu r(x_1^a)} | I_t^a = x]}{\mathbb{E}[e^{\mu r(x_1^a)} | I_t^a = x]} = b_{t,a+\mu}(x), \tag{33}$$

where the second equality is (7). Taking $n$ derivatives with respect to $\mu$ and evaluating at 0, we obtain

$$\frac{\partial^{n+1}}{\partial \mu^n \partial \nu} M(0, 0) = \frac{\partial^n}{\partial \mu^n} b_{t,a+\mu}(x)\big|_{\mu=0} = \frac{\partial^n}{\partial a^n} b_{t,a}(x). \tag{34}$$

The leftmost term is precisely the $(n+1)^{\text{th}}$ order joint cumulant. $\square$

**Proposition 4.** *(Implicit Tilt Matching.) Let $b_{t,a+h}$ be defined to all orders as in (14). Then*

$$\sum_{n>0} \frac{h^n}{n!} \frac{\partial^n}{\partial a^n}[b_{t,a}(x)] = \mathbb{E}[(e^{hr(x_1^a)} - 1)(\dot{I}_t^a - b_{t,a+h}(x)) | I_t^a = x] \tag{16}$$

*and $b_{t,a+h}$ is the minimizer of*

$$\mathcal{L}_{a \to a+h}^{\text{ITM}}(\hat{b}) := \int_0^1 \mathbb{E} \left\| \hat{b}_t(I_t^a) - T_{t,a,h}^{\text{ITM}} \right\|^2 dt, \tag{17}$$

*for any $h$, where expectation is taken over $(x_0, x_1^a) \sim \rho(x_0, x_1^a)$ conditional on $I_t^a = x$.*

*Proof.* Notice that the left side of (16) is equal to $b_{t,a+h}(x) - b_{t,a}(x)$ since the series contains all terms but $0^{\text{th}}$ order one in the Taylor series expansion in $h$ for $b_{t,a+h}(x)$. Next, we rewrite (19) for $c(x) = 1$

$$\mathbb{E}\left[ (b_{t,a+h}(x) - b_{t,a}(x)) + (e^{hr(x_1^a)} - 1)(b_{t,a+h}(x) - \dot{I}_t^a) \,\Big|\, I_t^a = x \right] = 0. \tag{35}$$

Rearranging, we obtain the following

$$b_{t,a+h}(x) - b_{t,a}(x) = \mathbb{E}\left[ (e^{hr(x_1^a)} - 1)(\dot{I}_t^a - b_{t,a+h}(x)) \,\Big|\, I_t^a = x \right], \tag{36}$$

which is the right side of (16). $\square$

**Proposition 5.** *(Variance control). Let $\mathcal{L}_{a \to a+h}^{\text{WFM}}$ and $\mathcal{L}_{a \to a+h}^{\text{ITM}}$ be the regression losses in (23) and (12). For sufficiently small $h$, the gradient estimator of WFM has variance at least as large as that of ACM:*

$$\text{Var}\left[ \nabla \mathcal{L}_{a \to a+h}^{\text{WFM}} \right] \geq \text{Var}\left[ \nabla \mathcal{L}_{a \to a+h}^{\text{ITM}} \right]. \tag{24}$$

*Proof.* The first variation (the Gateaux derivative) of the loss $\mathcal{L}_{a \to a+h}^{\text{c-ITM}}$ is

$$\delta\mathcal{L}_{a \to a+h}^{\text{c-ITM}}(\hat{b}) = 2 \int_0^1 \mathbb{E}\left[ c(I_t^a)(\hat{b}_t(I_t^a) - b_{t,a}(I_t^a)) + (e^{hr(x_1^a)} - c(I_t^a))(\hat{b}_t(I_t^a) - \dot{I}_t^a) \right] dt. \tag{37}$$

Therefore the Monte Carlo estimator used is

$$\xi_c := 2\Big( c(I_t^a)(\hat{b}_t(I_t^a) - b_{t,a}(I_t^a)) + (e^{hr(x_1^a)} - c(I_t^a))(\hat{b}_t(I_t^a) - \dot{I}_t^a) \Big) \tag{38}$$

where $t \sim \text{Unif}[0, 1]$. We will use the law of total variance

$$\text{Var}(\xi_c) = \mathbb{E}[\text{Var}(\xi_c | I_t^a)] + \text{Var}(\mathbb{E}[\xi_c | I_t^a]). \tag{39}$$

Notice that

$$\mathbb{E}[\xi_c | I_t^a] = \mathbb{E}\left[ e^{hr(x_1^a)}(\hat{b}_t(I_t^a) - \dot{I}_t^a) | I_t^a \right] \tag{40}$$

is independent of $c$ and therefore the same for any c-ITM variant. On the other hand, we have that

$$\text{Var}(\xi_c | I_t^a) = \text{Var}\left( e^{hr(x_1^a)}(\hat{b}_t(I_t^a) - \dot{I}_t^a) + c(I_t^a)\dot{I}_t^a | I_t^a \right). \tag{41}$$

Writing $e^{hr(x_1^a)} = 1 + \mathcal{O}(h)$, we see that

$$\mathbb{E}\left[\text{Var}(\xi_c|I_t^a)\right] = \mathbb{E}\left[\text{Var}\left((1 - c(I_t^a))\dot{I}_t^a|I_t^a\right)\right] + \mathcal{O}(h). \tag{42}$$

Recall that $\mathcal{L}^{\text{ITM}}_{a \to a+h}$ corresponds to taking $c(x) = 1$ and $\mathcal{L}^{\text{WFM}}_{a \to a+h}$ is $c(x) = 0$. When $c(x) = 1$, we have $\mathbb{E}\left[\text{Var}(\xi_c|I_t^a)\right] = \mathcal{O}(h)$. When $c(x) = 0$, we have $\mathbb{E}\left[\text{Var}(\xi_c|I_t^a)\right] = \mathbb{E}[\text{Var}(\dot{I}_t^a|I_t^a)] + \mathcal{O}(h)$. Provided that $\mathbb{E}[\text{Var}(\dot{I}_t^a|I_t^a)] > 0$, this completes the proof. We remark that $\mathbb{E}[\text{Var}(\dot{I}_t^a|I_t^a)] = 0$ occurs only for a very limited collection of couplings $\rho(x_0, x_1^a)$, such as the optimal transport coupling and would not be feasible in practice. Note that when $\hat{b}$ takes a parametric form, a similar proof holds.

$\square$

# B CONTROL VARIATES

In order to learn the optimal control variate, one may parameterize $c(x)$ as a small additional head or a standalone network and train it jointly with the velocity field to minimize the Monte Carlo variance of the ITM estimator. In particular, both objectives (20) and (21) can be used, where now we minimize these losses with respect to both the parameters of $\hat{b}_{t,a+h}$ and $c$. Jointly optimizing preserves the minimizer over $\hat{b}_{t,a+h}$ since the ITM objectives ensures that the unique minimizer is $b_{t,a+h}$ for any choice of $c$. By minimizing the loss with respect to the parameters of the $c$ network, we additionally minimize the variance of the objective.

# C EXPERIMENTS

## C.1 SAMPLING LENNARD-JONES POTENTIALS

The Lennard-Jones (LJ) potential is a widely used mathematical model that describes the potential energy between two neutral, non-bonding particles. This energy is calculated as a function of the distances between particles, capturing the balance between long-range attractive forces and short-range repulsive forces. It has the form

$$E^{\text{LJ}}(x) = \frac{\epsilon}{2\tau} \sum_{ij} \left(\left(\frac{r_m}{d_{ij}}\right)^6 - \left(\frac{r_m}{d_{ij}}\right)^{12}\right), \tag{43}$$

where $d_{ij} = \|x_i - x_j\|$ is the distance between particles $i$ and $j$, $\epsilon$ is the potential well depth, $r_m$ is the equilibrium distance at which the potential is minimized, and $\tau$ is the system temperature. We follow Köhler et al. (2020); Akhound-Sadegh et al. (2025) in adding a harmonic potential to the energy:

$$E^{\text{Total}}(x) = E^{\text{LJ}}(x) + \frac{1}{2}\sum_i \|x_i - \bar{x}\|^2, \tag{44}$$

where $\bar{x}$ is the center of mass of the system. We use the same parameters $\epsilon = 2.0, r_m = 1$ and $\tau = 1$ as Akhound-Sadegh et al. (2025) for our experiments. For the LJ-13 and LJ-55 datasets we use samples provided by the codebase in Akhound-Sadegh et al. (2025) which use the No-U-Turn-Sampler (NUTS) Hoffman & Gelman (2011).

In our experiments we use an EGNN Satorras et al. (2022). For LJ-13 we use three layers and 32 hidden dimensions which is approximately 45,000 parameters. For LJ-55 we use five layers and 128 hidden dimensions for a parameter count of approximately 580,000.

To compute the Effective Sample Size (ESS) we evaluate likelihoods $p_1(x_1)$ under our model $\hat{b}_t$ by

$$\log p_1(x_1) = \log p_0(x_0) - \int_0^1 \nabla \cdot \hat{b}_t(x_t) dt \tag{45}$$

to compute importance weights $w(x_1) = \frac{\rho_{1,a=0}(x_1)e^{r(x_1)}}{p_1(x_1)}$ and then compute the ESS as

$$\text{ESS} = \frac{(\sum_{i=1}^N w_i)^2}{N \sum_{i=1}^N w_i^2}. \tag{46}$$

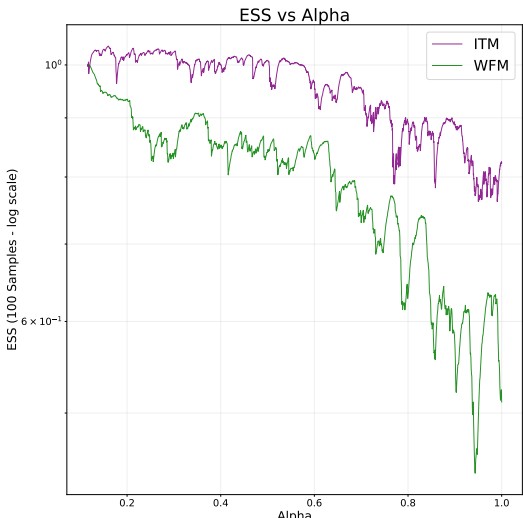

**Figure 5:** ESS evolution with alpha for ITM and WFM.

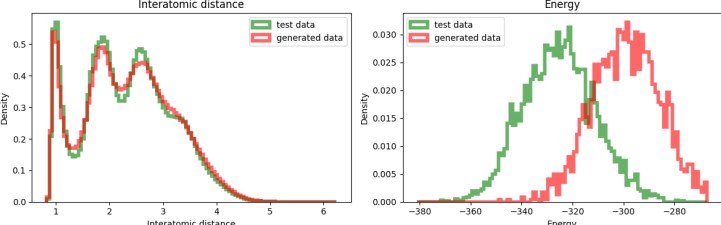

**Figure 6:** Histograms of interatomic distance and Energy on LJ55.

For ETM we use the ESS to dynamically update the step size $h$ for transitions from $\rho_{1,a}$ to $\rho_{1,a+h}$. If the ESS drops below a given threshold, we decrease the step size to $h' = 0.5h$ and attempt the transition from $\rho_{1,a}$ to $\rho_{1,a+h'}$. For ITM we use a fixed step size of $h = 0.001$. We use $800$ gradient steps per anneal update. We use a simple Euler integrator with $100$ steps in each case. We use the linear interpolant $I_t = (1-t)x_0 + tx_1$ for our experiments.

## C.2 PLOTS

We include a plot comparing the evolution of the 100 sample ESS for an ITM and WFM run in Figure 5. For the Lj55 experiement, we also include a histogram of interatomic distances amongst particles in the system and a histogram of the energy of 10000 samples of the system in Figure 6.

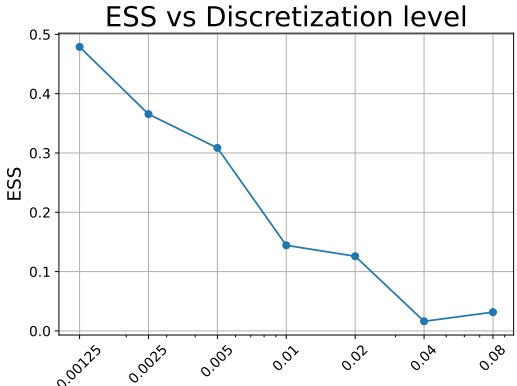

**Figure 7:** ITM performance scaling with discretization step size $h$.

---

**Algorithm 1:** Tilt Matching (TM)

---

**Input:** Pretrained drift $b_{t,0}$; reward $r(x)$; annealing schedule $\{a_k\}_{k=0}^{K}$ with steps $h_k = a_{k+1} - a_k$;
interpolant $I_t^a$ (linear shown); epochs $E$; batch size $B$, ETM or ITM.
**Output:** Tilted drift $b_{t,1}$.
**for** $k = 0, \ldots, K-1$ **do**
    // Current model is $b_{t,a_k}$; goal is $b_{t,a_{k+1}}$
    Initialize $\hat{b}_t \leftarrow b_{t,a_k}$
    **for** $epoch = 1, \ldots, E$ **do**
        Draw $B$ samples $(x_0, x_1^{a_k})$ with $x_1^{a_k} \sim \rho_{1,a_k}$ (from model or buffer), $t \sim \text{Unif}[0,1]$.
        $I_t^{a_k} \leftarrow (1-t)x_0 + tx_1^{a_k}$;    $\dot{I}_t^{a_k} \rightarrow x_1^{a_k} - x_0$
        **if** *ETM* **then**
            $T_{t,a_k,h_k} \leftarrow b_{t,a_k}(I_t^{a_k}) + h_k(\dot{I}_t^{a_k} r(x_1^{a_k}) - b_{t,a}(I_t^{a_k}))r(x_1^{a_k})$
        **if** *ITM* **then**
            $T_{t,a_k,h_k} \leftarrow b_{t,a_k}(I_t^{a_k}) + \left(e^{h_k r(x_1^{a_k})} - 1\right)\left(\hat{b}_t(I_t^{a_k}) - \dot{I}_t^{a_k}\right)$
        $\mathcal{L} \leftarrow \frac{1}{B}\sum^B \|\hat{b}_t(I_t^{a_k}) - T_{t,a_k,h_k}\|^2$
        Update the parameters of $\hat{b}_t$ by gradient descent to minimize the TM loss.
    Set $b_{t,a_{k+1}} \leftarrow \hat{b}_t$.
**return** $b_{t,1}$.

---

## D    EXAMPLE IMAGES FROM FINE-TUNING EXPERIMENTS

In this appendix, we display a random selection of images from the base model and the model fine-tuned under the Euler Tilt Matching objective. It can be seen that images generated from the fine-tuned model better adhere to the given text prompt, which aligns with the numerical results in the main text.

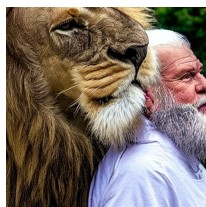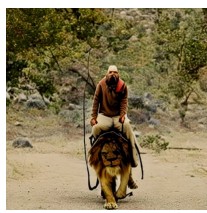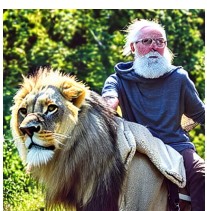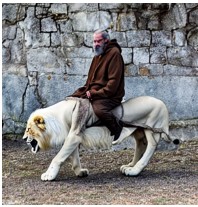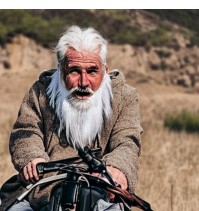

**Figure 8:** Images generated from the base model with prompt: *old man ( long white beard and a hood ) riding on lions back*

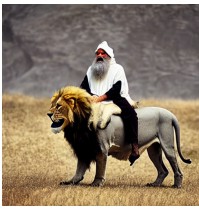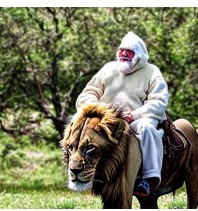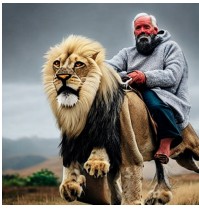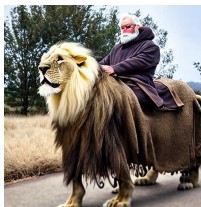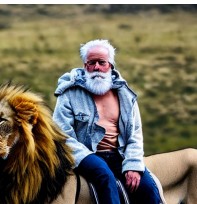

**Figure 9:** Images generated from the Tilt Matching fine-tuned model with prompt: *old man ( long white beard and a hood ) riding on lions back*

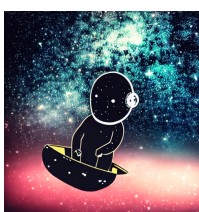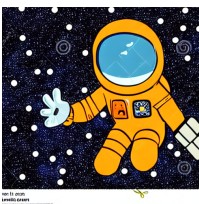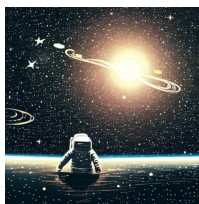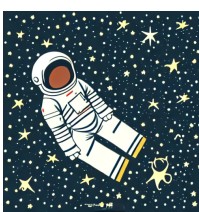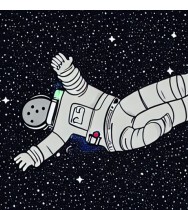

**Figure 10:** Images generated from the base model with prompt: *astronaut drifting afloat in space, in the darkness away from anyone else, alone, black background dotted with stars, realistic*

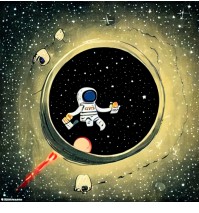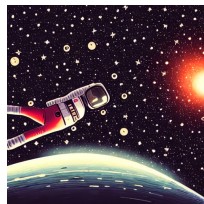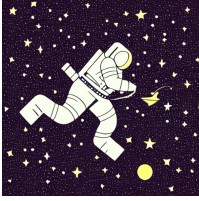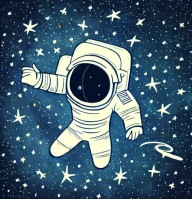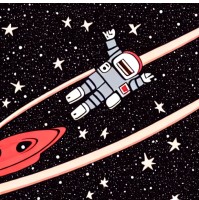

**Figure 11:** Images generated from the Tilt Matching fine-tuned model with prompt: *astronaut drifting afloat in space, in the darkness away from anyone else, alone, black background dotted with stars, realistic*

## LLM USAGE

In preparing this paper, we used large language models (LLMs) as assistive tools. Specifically, LLMs were used for (i) editing and polishing the text for clarity and readability, and (ii) formatting matplotlib code. The authors take full responsibility for the content of this paper.

