# OpenReview forum: "Tilt matching for scalable sampling and fine-tuning"
_ICLR.cc/2026/Conference — Submitted to ICLR 2026_

### Official Review · Reviewer_PMF8 · 2025-10-17

**Soundness:** 3
**Presentation:** 3
**Contribution:** 3
**Rating:** 2
**Confidence:** 4

**Summary:**

The paper proposes a method to adapt flow matching models for sampling from tilted distributions. Starting from a flow matching model that samples from $\rho_1$, the goal is to generate samples from $\rho_{1,a}(x) \propto \rho_1(x) \exp(a, r(x))$ where $r$ is a reward function and $a \in [0,1]$ controls the tilt. This allows sampling from unnormalized densities and performing reward-based fine-tuning. The authors derive the optimal velocity field transporting to $\rho_{1,a}$ and its derivative with respect to $a$. Using a Taylor expansion in $a$, they obtain mean-square objectives whose minimizers approximate the optimal velocity fields. The first-order approximation defines Explicit Tilt Matching (ETM), while higher-order corrections lead to Implicit Tilt Matching (ITM), which requires a fixed-point solution. They show that an importance-weighted version of the classical flow matching loss corresponds to a special case of the ITM loss with a control variate. For small time steps, this makes the importance-weighted loss higher in variance, highlighting the stability of ITM. Experiments on (i) Lennard-Jones sampling (from the iDEM paper) and (ii) diffusion fine-tuning (from the Adjoint Matching paper) confirm that ETM and ITM achieve superior performance, demonstrating scalability and robustness in both sampling and fine-tuning settings.

**Strengths:**

* The paper presents a very elegant and well-formulated idea.
* The paper acknowledges its limitations (e.g., approximation error due to incomplete minimization of the objective, discretization error) and proposes reasonable solutions to address them

**Weaknesses:**

* The paper contains numerous typographical errors.
* The paper does not mention or discuss the closely related work [1], which introduces a very similar idea referred to as "temperature guidance". This work was released contemporaneously with Adjoint Sampling and PITA.
* The experimental evaluation of the sampling component is weak, as it compares only to neural samplers and omits well-established sampling techniques such as (Adaptive) Parallel Tempering [2] or (Adaptive) Sequential Monte Carlo [3,4]. Moreover, several baselines used for comparison (e.g., DDS, PIS, iDEM) are already known to perform poorly, as reported in prior work [5,6].
* There are no ablation studies on simple Gaussian mixture models to analyze the behavior and properties of the proposed sampling algorithm (see [5,7] for examples in sampling).
* The paper does not include an empirical comparison with the importance-weighted variant (WFM).
* There is no clear description of key experimental details, including training hyperparameters, time discretization (with respect to $t$ or $a$), ODE/likelihood integration method, noising schedule, number of samples used, standard errors for the sampling experiments, or any ablation on computational cost.
* The fine-tuning metrics reported in Table 2 do not demonstrate a clear improvement with the proposed method, particularly in terms of the ClipScore, which appears largely unchanged.

[1] Rissanen, S., Ouyang, R., He, J., Chen, W., Heinonen, M., Solin, A., & Hernández-Lobato, J. (2025). Progressive Tempering Sampler with Diffusion. In Proceedings of the 42nd International Conference on Machine Learning (pp. 51724–51746). PMLR.

[2] Syed, S., Bouchard-Côté, A., Deligiannidis, G., & Doucet, A. (2021). Non-Reversible Parallel Tempering: A Scalable Highly Parallel MCMC Scheme. Journal of the Royal Statistical Society Series B: Statistical Methodology, 84(2), 321-350.

[3] Saifuddin Syed, Alexandre Bouchard-Côté, Kevin Chern, & Arnaud Doucet. (2024). Optimised Annealed Sequential Monte Carlo Samplers.

[4] Yan Zhou, Adam M. Johansen, & John A.D. Aston (2016). Toward Automatic Model Comparison: An Adaptive Sequential Monte Carlo Approach. Journal of Computational and Graphical Statistics, 25(3), 701–726.

[5] Blessing, D., Jia, X., Esslinger, J., Vargas, F., & Neumann, G. (2024). Beyond ELBOs: A Large-Scale Evaluation of Variational Methods for Sampling. In Proceedings of the 41st International Conference on Machine Learning (pp. 4205–4229). PMLR.

[6] Maxence Noble, Louis Grenioux, Marylou Gabrie, & Alain Oliviero Durmus (2025). Learned Reference-based Diffusion Sampler for multi-modal distributions. In The Thirteenth International Conference on Learning Representations.

[7] Grenioux, L., Noble, M., & Gabrié, M. (2025). Improving the evaluation of samplers on multi-modal targets. In Frontiers in Probabilistic Inference: Learning meets Sampling.

**Questions:**

* In the paragraph "Approximation error due to incomplete minimization of the objective", you mention using MCMC steps. How sensitive is the method to the number steps? What is their computational cost relative to the overall procedure?
* Regarding the sampling procedure, a more advanced version of the WFM approach is proposed in [8], where importance sampling is replaced by an Independent Metropolis–Hastings (IMH) scheme. This method follows the same principle as yours (progressively training at decreasing temperatures) but without the clever use of the $a$-derivative of the velocity field. Instead, it reuses the network trained at step $a$ as a warm start for step $a+h$. **(a)** Could you provide an empirical comparison against WFM (IS + FM) and MFM [8] (IMH + FM)? **(b)** Could you perform an ablation study to demonstrate the benefit of incorporating the $a$-derivative, e.g., by comparing your current strategy (with MCMC refreshments) against a variant that simply reuses the previous network as a warm start for the next step?
* The energy interpolation path defined in Eq. (24) is known to suffer from mode-switching issues (see, e.g., [9]), meaning that the relative proportions of different low-energy regions can change significantly and abruptly as $a$ varies. Such rapid shifts may invalidate the Taylor approximations and reduce the reusability of the velocity field learned at step $a$ for step $a+h$. How do you address or mitigate this potential instability in your approach?

[8] Cabezas, A., Sharrock, L., & Nemeth, C. (2024). Markovian Flow Matching: Accelerating MCMC with Continuous Normalizing Flows. In Advances in Neural Information Processing Systems (pp. 104383–104411). Curran Associates, Inc.

[9] Bálint Máté, & François Fleuret (2023). Learning Interpolations between Boltzmann Densities. Transactions on Machine Learning Research.

---

> ### Author Response · Authors · 2025-11-27
>
> Thank you for your review. We address your concerns below.
>
>
> ### Weaknesses
>
> **Typographical errors**
>
> We apologize for the typographical errors. Please see that we have taken care to fix as many mistakes as we can in the revised manuscript.
>
>
> **Missed citations**
>
> We apologize for having missed "Progressive Tempering Sampler with Diffusion". PTSD defines a temperature annealing path and sequentially trains separate diffusion models along this discrete ladder. Leveraging models trained at higher temperatures $T_1$ and $T_2$, the method employs a finite difference approximation with respect to temperature to extrapolate the score function at a lower target temperature $T_3$. This derived score is used to generate approximate samples at the new temperature $T_3$. To ameliorate the sample quality and correct for approximation bias, the algorithm applies truncated importance resampling followed by a local parallel tempering refinement between the adjacent temperature levels. Finally, the diffusion models are fine-tuned on these refined sample buffers. Tilt Matching employs a similar strategy of iteratively fine-tuning models along a (temperature) annealing path. However, ETM derives a different update rule for the drift using the covariance ODE relation instead of a finite difference approximation. ITM derives an update rule that is unbiased provided that the previous drift is correctly learned. Note that even if the score functions are perfectly learned at $T_1$ and $T_2$ for PTSD, the finite difference approximation still introduces an error for the approximate score function at $T_3$. The technique of using the parallel tempering refinement is compatible with Tilt Matching while avoiding the finite difference approximation error. We appreciate the reviewer for having brought this to our attention.
>
> We have included a discussion in the related works section in the revised document.
>
>
> **Comparison to non-neural samplers**
>
> We agree that our comparisons are only with neural samplers. The field of neural sampling is still trying to match the performance of many dedicated sampling methods. Therefore we felt the correct comparison was with other neural samplers, something that is a common practice for papers in this area. However, we are open to including comparisons with other methods if the reviewer believe this would improve the paper.
>
> **Ablations on GMMs**
>
> We did not provide results for the Gaussian mixture models as our results on the LJ systems already shows rigourously that our sampling method can produce unbiased samples from unnormalized densities and is more indidcative of real-world tasks where one may want to employ neural samplers.
>
> **Comparison with WFM**
>
> We would like to clarify that we are not claiming that weighted flow matching (WFM) is a poor or incorrect algorithm. In fact, WFM is mathematically a special case of our framework, obtained by choosing the control variate $c=0$. Our theoretical discussion of WFM was included solely to motivate the broader family of ITM objectives and to highlight why other instantiations can enjoy improved stability. The higher variance of WFM implies that, in practice, it typically requires smaller learning rates to remain stable. For example, we found that WFM could diverge if the learning rate was not decreased within each annealing step from $a$ to $a+h$, rendering the resultant sampler useless. By contrast, ITM remains stable under larger step sizes, making it a more robust method. However, with a well-tuned learning rate schedule, both methods perform well. The higher variance objective simply means that training is less stable.
>
> **Experimental details**
>
> We have included a more detailed description of experimental details in the revised paper, please see Appendix C.
>
> In fine-tuning experiments, a common issue is "reward hacking" where optimizing solely for a single reward leads to degradation in other perceptual quality metrics. Our purpose in Table 2 is to show precisely that this does not occur: while we train *only* on ImageReward, other quality metrics remain stable or improve over baseline. We will clarify this intent in the paper, as the current text can be misread as reporting minimal improvements rather than demonstrating robustness against reward hacking. We also now include ImageReward scores. Our method achieves higher image reward at the same level of tilt as compared to the previous state-of-the-art in adjoint matching.

---

> > ### Author Response · Authors · 2025-11-27
> >
> > ### Questions
> >
> > **Question on MCMC steps**
> >
> > The method is robust to the number of steps for MCMC. For our experiments, the number of calls to the energy functions is the same order of magnitude as the number of calls to the neural network.
> >
> >
> > **Comparison to other methods**
> >
> > Markovian Flow Matching (MFM) differs fundamentally from our WFM/ITM objectives. MFM uses a combination of local and non-local MCMC kernels to approximately transport samples from $p_{1,a}$ to $p_{1, a+h}$, and then treats the resulting particles as if they were exact samples from $p_{1, a+h}$ when applying flow matching, which makes the objective biased. By contrast, WFM and ITM never assume an exact transport of samples; instead, they propagate the training objective across $a$ via exact reweighting identities, giving unbiased estimators of the tilted drift. Unfortunately, MFM does not report results on Lennard–Jones potentials and would require substantial new implementation in our setting, so we do not include it as a baseline in this revision.
> >
> >
> > **Question on energy interpolation path**
> >
> > We agree that the energy interpolation path in Eq. (24) can suffer from mode-switching issues, as documented in prior work. However, this interpolation path is *only* used to define the tilted target densities $p_{1,a}$. A method that attempted to transport mass along this energy path would indeed inherit its mode-switching instabilities.
> >
> > Our approach does *not* transport along the energy interpolation path. Indeed, this is the point of the proposed approach. Instead, the drift $b_{t,a}$ always transports samples along the stochastic interpolant path, which is the fixed linear interpolation from the Gaussian prior $p_0$ to the current target $p_{1,a}$. This path is well known to have favorable regularity and does not exhibit the mode-switching behavior described. Thus, although the tilted densities $p_{1,a}$ may change sharply with $a$, the flow we learn never transports along the problematic energy path, and therefore does not suffer from mode-switching instabilities.
> >
> > Moreover, the Covariance ODE shows that the drift $b_{t,a}$ changes continuously in the annealing parameter $a$ under mild assumptions.
> >
> >
> > Please let us know if you find these revisions suitable. If not, we are happy to continue amending the paper accordingly. If these have addressed at least some of your concerns, we would appreciate any adjustment in score you see fit. :)

---

> > > ### Comment · Reviewer_PMF8 · 2025-11-28
> > > **Answer to the rebuttal**
> > >
> > > I thank the authors for the detailed rebuttal addressed to me and the other reviewers.
> > >
> > > **On related work (PTSD):**
> > > Thank you for adding a discussion of PTSD. I agree that it is an important and closely related approach, and I believe it would be valuable to include an experimental comparison to further contextualize the advantages of ETM/ITM.
> > >
> > > **On comparisons with classical samplers:**
> > > While I understand the rationale for comparing primarily against neural samplers, I align with the other reviewers in considering comparisons to established non-neural samplers [2,3,4] essential. These methods remain strong baselines for challenging multi-modal distributions and unnormalized densities. Without such comparisons, it is difficult to assess whether the proposed method provides genuine practical benefits beyond the neural-sampling literature. As such, I cannot raise my score without this evaluation.
> > >
> > > **On the absence of Gaussian mixture ablations:**
> > > I disagree with the claim that Lennard–Jones results alone “rigorously” demonstrate unbiased sampling. In fact, Fig. 6 shows notable discrepancies between the sampled and true energy distributions, and Table 1 indicates low ESS, suggesting non-negligible bias. Moreover, as the community has repeatedly emphasized, Lennard–Jones benchmarks offer limited diagnostic power: they do not reveal whether all modes are discovered or sampled in the correct proportions. Simple synthetic multi-modal targets such as Gaussian mixtures are now a standard requirement for validating neural samplers (see for instance [7] and also raised by another reviewer). These experiments are inexpensive, interpretable, and highly informative. I therefore maintain that an ablation on Gaussian mixtures is necessary for a robust evaluation, and I cannot increase my score without it.
> > >
> > > **On MFM:**
> > > My comment on MFM was similar to the one on PTSD: it is a closely related method whose objective differs only slightly. Even if full reproduction is difficult for LJ experiments, some comparison on simpler settings would strengthen the paper.
> > >
> > > **On potential mode-switching across tilt parameters:**
> > > Your response acknowledges that the tilted densities $p_{1,a}$ may change sharply with $a$. This is precisely my concern: if the relative mass across modes shifts abruptly as $a$ varies, then the drift $b_{t,a}$ learned at one tilt level may be poorly suited as an initialization for $b_{t,a+h}$, causing instabilities or slow adaptation. The argument based on the covariance ODE does not fully resolve this issue, since the expectations involved can themselves vary dramatically when the distribution changes its modal structure. This remains, in my view, an important limitation of the method.
> > >
> > > Overall, I appreciate the authors’ effort to improve the manuscript and address the concerns raised. However, the experimental shortcomings remain significant: (i) the lack of any comparison with established classical samplers, and (ii) the absence of an evaluation on Gaussian mixture models.
> > >
> > > These issues prevent me from updating my score. I therefore continue to recommend rejection.

---

### Official Review · Reviewer_ojYB · 2025-10-31

**Soundness:** 3
**Presentation:** 2
**Contribution:** 3
**Rating:** 4
**Confidence:** 4

**Summary:**

The paper introduces a theoretically novel approach, tilt matching. The analysis is interesting and different from prior work, and the proofs appear sound. Empirically, the authors conduct experiments in the traditional energy function land and also in the text-to-image generation land.

**Strengths:**

There are two main streams for obtaining the posterior distribution from the prior and the reward function.  AM like approach requires the reward to be differentiable while other approaches do not.  The latter approaches typically fall into the importance sampling land which is traditionally be known for having poor sample efficiency like iDEM. The proposed approach improved the sample efficiency compared to other importance sampling baselines which offer an alternative approach when dealing with non-differentiable reward.

In addition, the theory behind tile matching is also interesting as it formulates the changes needed as a regression objective which we have known from FM/DM that it is stable.

**Weaknesses:**

While the paper’s theoretical perspective is sound, the empirical results are less compelling than expected.
Table 1 reports higher ESS than iDEM and substantially better performance, but a strong baseline, ASBS is not included. Assuming comparable experimental settings, ASBS achieves better results on the more complex energy function (LJ-55), which raises questions about the scalability of the proposed approach.

For Table 2: if you followed the same experimental setup as Domingo-Enrich et al. (2025), did you train and test on the same set of 100 prompts? Although ETM marginally outperforms AM, the diversity score drops considerably. This may be due to how you sample $X_1$, the ODE/SDE noise schedule, etc. In addition, there is no comparison of computational cost with AM. I suspect your approach is more expensive; reporting number of samples, number of function evaluations would be valuable to the community.

Please also check the question section.

[1] Liu, Guan-Horng, et al. "Adjoint Schr\" odinger Bridge Sampler." arXiv preprint arXiv:2506.22565 (2025).

**Questions:**

- The derivation of equation (26) is missing. How did you arrive at (26) from the tilted distribution defined at X_1?
- Table 2, Looking at the results reported in Table 2 for AM, the default scheduler’s results are used compared to the fine-tuning scheduler.  Also, why did the author choose to omit the ImageReward metric?
- Importantly, can the author compare the sample efficiency in Table 2? How many samples do you need to generate from iterative denoising to  arrive at the results compared to AM?
- Could the author explain the purpose of flow matching recentering in the algorithm and provide a theoretical justification? “A few flow matching steps” sounds vague and makes the algorithm hard to reproduce.

---

> ### Author Response · Authors · 2025-11-27
>
> Thank you for your thoughtful feedback. We address your questions below.
>
> ### Weaknesses
>
> **Comparison with ASBS**
>
> We apologize for having missed ASBS in our table, which we have included in the updated manuscript. Please note that ITM is within the error bar for the energy $W_2$, but improves upon ASBS on the distance $W_2$. We are happy to iterate on these comparisons if useful. If you know of code that reproduces the ASBS $W_2$ computation, we are happy to use it for a direct comparison as well, as ours relies on the open-source code provided by iDEM and PITA for fair comparisont there.
>
> **Comparison of setup with AM**
>
> We followed the same experimental setup and used the same prompts as Domingo-Enrich et al. (2025). For Table 2, we use the finetuning scheduler for training and evaluation, setting eta = 0 for inference (deterministic ODE sampler) to align with the prior work.
>
> **Diversity score**
>
> While the diversity score did drop, we would like to mention that it is unclear whether this is desirable or not. By tilting the target distribution by ImageReward, one aims to concentrate samples in areas of high reward. Therefore, the tilted distribution likely should have less diversity.
>
> **Computational cost**
>
> Comparing computational cost is unfortunately difficult, since the two methods have substantially different requirements. Per gradient step, Adjoint Matching requires differentiating through the reward function and evaluating the network along a trajectory, which is significantly more expensive than a single Tilt Matching update. However, this also means Tilt Matching typically requires more gradient updates. A direct wall-clock comparison is highly implementation-dependent, but in our setup the two runtimes were about the same.
>
>
> ### Questions:
>
> **Derivation of equation (26) (Question 1)**
>
> Thanks for pointing this out. We apologize for having missed the derivation of Eq. (26). We have included it in the revised paper (see Appendix).
>
> **Table 2 results (Question 2)**
>
> We apologize for having missed this in our original paper. We have updated our table to include both changes. As you can see, for the same tilt, our method achieves a higher ImageReward score than Adjoint Matching.
>
> **Sample efficiency (Question 3)**
>
> In terms of sample efficiency, it is unfortunately again difficult to compare the two methods in a fair way. Adjoint Matching uses gradients of the reward and entire trajectories, which means that AM can rely on fewer samples than TM; however, each sample is substantially more expensive to store and acquire, as it relies on these reward gradients and solving a lean adjoint equation. We find that our approach takes a few hours on 4 A100s to do.
>
> **Flow matching steps (Question 4)**
>
> We acknowledge that our description of "refreshing" was unclear. In early experiments, we periodically generated samples from the current drift $\hat b_{t,a}$, applied a small number of MALA refinement steps targetting $p_{1, a}$, and then performed a few standard flow matching updates on these refined samples to update $\hat b_{t,a}$. This mechanism was never used in our Stable Diffusion experiments, and after further experimentation we removed it entirely from our current results. We have removed references to it from the manuscript to avoid confusion.
>
>  We have updated the text accordingly and would appreciate any requisite adjustment in score. :)

---

### Official Review · Reviewer_QPg2 · 2025-10-31

**Soundness:** 4
**Presentation:** 4
**Contribution:** 2
**Rating:** 4
**Confidence:** 4

**Summary:**

The paper introduces a method to tilt stochastic interpolants in the objective either to sample from a target distribution known up to normalization or to fine-tune the model according to a given reward function.

The proposed method is iterative, progressively annealing the strength of the titling from an untilted model up to the desired reward function/unormalized target distribution. The idea is to rely on the predicted evolution of the optimal velocity field along this annealing of the tilting parameter to derive a loss for the velocity field at tilt a+h (where h is the increment step) when the velocity has been learned at tilt a. This derivative can be estimated using covariances and yields losses that can be estimated using generated samples at previous values of the tilt. Two versions of the algorithm are discussed according to different strategies to discretize the evolution of the velocity field along the variation of tilt a.

A variance reduction method is also proposed that allows to reduce the variance of the flow matching objective, but it is unclear if it is implemented in the numerical experiments.

Numerical experiments are presented on:
-  sampling from the Boltzmann distribution of clusters of Lennard-Jones particles of either 13 or 55 particles where an increase of performance seems to be achieved compared to other samplers using continuous time generative models.
- and fine-tuning of Stable-Diffusion using the ImageReward score where the performance seem to be competitive with adjoint matching, in particular without the need to fine-tune the strength of a reward multiplier.

**Strengths:**

- 1 - The paper is overall well written and easy to follow.
- 2 - The proposed approach is to my knowledge entirely novel and the numerical experiments are limited but encouraging.

**Weaknesses:**

- 3 - The method requires retraining iteratively to a satisfactory accuracy as the training objective at tilt a+h leverages samples from the model at tilt a: This entails a rather important computational budget and the methods proposed by the authors to “refresh” the model to circumvent this issue are not entirely clear.
- 4 - The second main limitation of the paper is the limited numerical evaluation of the proposed approach as no systematic studies are conducted:
	- On the effect of the discretization step of the annealing is the tilt, while this governs the computational cost of the method. While the ITM is supposedly free of “discretization error” it seems unlikely that one can go in one step from a=0 to a=1. This discussion is currently missing in the paper.
	- For sampling from unnormalized densities, the paper produces benchmarks that are usual but that are providing little information on the expected behavior of the method beyond these benchmarks. I note that for LJ55, nor the ESS, nor the histograms of energies and pairwise distances are provided. As multimodality of the target distribution and high-dimensionality are two big challenges in sampling that justify going past traditional MCMC or Sequential Monte Carlo methods, systematic experiments of increasing dimension and increasing separation of the modes are simple to run and offer crucial information to assess the method (see e.g. Grenioux et al 2025).
	- It is unclear whether the variance reduction approach proposed is employed and how in the experiments.


Minor:
- Figures are not referenced in the main text.
- line 410, typo, Tilt Madtching.
- Some sentences in the introduction can be made more precise:
	- L35 “These models work by building a continuous time map connecting a base distribution to a target distribution, realized by solving a differential equation whose coefficients are neural networks.” I am not sure that we should say that the coefficients are neural networks rather than terms themselves of the differential equations involve neural networks.
	- L078 “where we achieve state-of-the-art performance on sampling Lennard-Jones potentials”, the authors should add “with diffusion based samplers”, since for these simple LJ clusters at relatively high-temperature it is unclear that classical samplers are not more efficient in wall-clock time.

Ref:
Grenioux, Louis, Maxence Noble, and Marylou Gabrié. “Improving the Evaluation of Samplers on Multi-Modal Targets.” Paper presented at Frontiers in Probabilistic Inference: Learning meets Sampling. ICLR Workshop on Frontiers in Probabilistic Inference: Learning Meets Sampling, April 24, 2025. https://openreview.net/forum?id=d91E9RhVFU.

**Questions:**

- 5 - Related to 3, can the authors comment on the computational budget of their approach compared to competitors in the experiments they ran?
- 6 - Related to 4, Can the authors clarify their refresh procedure? On which newly generated samples would they perform flow matching? Which samples would a buffer contain? Would the refined steps be run with a local MCMC steps? What would be the limitation of the approach?
- 7 - Related to 4, for LJ55, although the ESS is computationally costly to compute, the histograms of energies and pairwise distances only necessitate generating a batch of samples, can you please report them?
- 8 - line 269 - Which ESS exactly would be computed to dynamically decide for the update step in explicit tilt matching? ESS in sampling at step a+h with model trained at tilt a?
- 9 - In loss (17), can the authors clarify if the residual is evaluated using the estimated field $\hat b_t$ where $b_{t,a+h}$ lies in the definition of residual?
- 10 - How exactly would the control variate function be learned? Is it used in the presented experiments?

---

> ### Author Response · Authors · 2025-11-27
>
> Thank you for your thoughtful feedback. We address your questions below.
>
> ### Computational cost (Weakness 3 & Question 5)
>
> Our goal in our experiments was to demonstrate that, if one wants a sampler as accurate as possible, there is no inherent bottleneck in our method: simply decreasing the step size or increasing the amount of training yields unbiased samples to any desired precision. For ITM we used a step size of 0.001 and trained for 800 steps per update, for a total of 800,000 training steps. For a more computationally light variant, we ran ITM on LJ13 with a step size of 0.01 and trained for 800 steps per update, for a total of 80,000 training steps. The final results are: energy $W_2 = 2.52$, distance $W_2 = 0.012$ and $ESS = 0.346$. For comparison, PITA performs 250,000 steps in training and is more expensive per step that ITM. As such, we are more efficient than the alternative state-of-the-art of neural samplers, with better performance. In addition, the current SOTA algorithm has a bias in how the importance weights it needs are computed, which our method avoids.
>
>
> ### "Refresh" procedure (Weakness 3 & Question 6)
>
> Thanks for pointing this out. We acknowledge that our description of "refreshing" was unclear. In early experiments, we periodically generated samples from the current drift $\hat b_{t,a}$, applied a small number of MALA refinement steps targetting $p_{1, a}$, and then performed a few standard flow matching updates on these refined samples to update $\hat b_{t,a}$. This mechanism was never used in our Stable Diffusion experiments, and **after further experimentation** we removed it entirely from our current results. We have removed references of it in the updated manuscript to avoid confusion.
>
> ### Energies and pairwise distance for LJ55 (Question 7)
>
> Thanks for the suggestion. We have added the energies and pairwise distances comparison plot as Figure 6 in the appendix, and can incorporate it into the main text upon your approval.
>
>
>
> ### ETM $\alpha$ discretization (Question 8)
>
> We agree that the paper should more clearly explain the discretization used by ETM. We used an ESS-based adaptive step size, where the ESS is the standard importance-sampling ESS targeting $p_{1, a+h}$ with the current model $\hat b_{t,a+h}$ as given by Eq. (45) and (46). If the ESS is below a threshold, we decrease the step size $h \to h'$, and then attempt to anneal from  $p_{1, a}$ to $p_{1, a+h'}$. ITM has no discretization bias, so we simply chose a fixed step size of 0.001 for all experiments. This information about ETM is now provided in the experimental details in the appendix.
>
> ### Role of $b_{t,a+h}$ in residual (Question 9)
>
> Regarding Loss (17), the residual is computed using the current estimate $\hat  b_{t, a+h}$ of $b_{t,a+h}$. Our revised manuscript makes this clearer and includes a more thorough discussion of the ITM gradient objectives.
>
> ### Variance reduction (Weakness 4 & Question 10)
>
> In all experiments, we used the simple choice $c(x)=1$, which is close to optimal for small $h$ and avoids introducing extra trainable components. We did not use learned control variates in the current results, and we will clarify this in the revision.
>
> The Appendix now includes a more detailed explanation of how a learned control variate could be implemented. In summary, one may parameterize $c(x)$ as a small additional head and train it jointly with the velocity field to minimize the Monte Carlo variance of the ITM estimator. Its parameters can be updated using the same samples used for the main ITM loss, requiring no additional sampling and adding minimal overhead.
>
> We also thank the reviewer for highlighting minor corrections. We have updated the text accordingly and would appreciate any requisite adjustment in score. :)

---

> > ### Comment · Reviewer_QPg2 · 2025-11-27
> >
> > I thank the authors for their answers to my questions. If they answered all the other points, they did not address my major concern for simple more systematic experiments.
> >
> > 1) Concerning their answer to the question of computation costs, the authors claim: "Our goal in our experiments was to demonstrate that, if one wants a sampler as accurate as possible, there is no inherent bottleneck in our method: simply decreasing the step size or increasing the amount of training yields unbiased samples to any desired precision."
> > While the numbers reported by the authors in their answer are interesting and worthy of discussion in the paper,
> > I do not see in the paper an experiment where, for a fixed task, the discretization is systematically decreased and the accuracy is shown to systematically improve.
> >
> > 2) In their global reply, the authors also argue that a systematic experiment of Gaussian mixture is unnecessary and make no mention of this aspect in their individual answer to my review. I disagree with their view. Unlike LJ clusters, a systematic experiment on Gaussian mixtures allows to systematically increase the complexity of the task in two clearly identifiable sources of difficulty: dimension and separation of the modes. Given the multiplication of papers in this area, such systematic benchmarks would allow to clearly compare algorithms and understand their strengths and weaknesses.
> >
> > Provided that (i) unless I am misunderstanding something, both of these tests are straightforward to run and (ii) this line of research crucially lacks systematic assessment, I am not willing to raise my score if the authors do not address my concern.

---

> > > ### Author Response · Authors · 2025-12-03
> > >
> > > To address the reviewer's concerns we include Fig. 7 in Appendix C. We plot the ESS vs discretization level where, for each annealing step, we perform 800 training steps. Therefore as the discretization level decreases, total training time increases. We can see a clear improvement in the ESS with discretization level or total compute.

---

### Official Review · Reviewer_dbLF · 2025-10-31

**Soundness:** 3
**Presentation:** 3
**Contribution:** 2
**Rating:** 2
**Confidence:** 4

**Summary:**

This paper introduces Tilt Matching, a framework for: sampling from unnormalized densities and fine-tuning large-scale generative models based on reward functions. The core contribution is the derivation of a "Covariance ODE": a dynamical equation that describes the evolution of the stochastic interpolant's velocity field with respect to the tilting parameter. Two variants ETM and ITM are proposed to learn the parameterized models for tilting. A key advantage of the proposed methodis that it doesn’t require gradients of the reward function or backpropagation through the flow's trajectory.

**Strengths:**

- Writing: I found the writing to be easy to follow, though the work assumes familiarity with a lot of background material.
- Theoretical Insights: The derivation of the Covariance ODE is very interesting and the connection to the Esscher transform is quite insightful as well.
- Promise of practical Advantages: Proposed ETM/ITM hold the promise to solve key problems with existing methods: 1) avoid backprop through trajectories, 2) avoid reward gradients, and 3) lower variance than WTM. The empirical validation starts to offer some evidence in this direction.

**Weaknesses:**

- Disconnect between theory and empirical validation: Paper offers ITM as a superior alternative to ETM. However, ETM seems to outperform ITM on the tested small scale configuration for sampling as well as in term of stability for finetuning (no ITM results are reported due to it’s instability in finetuning experiments). Given that the paper spends a significant amount of space motivating and deriving the ITM variant, I find the empirical validation preliminary and lacking. Trusting Table-1, I am willing to believe that ITM is more scalable, however rest of the experiments fail to establish it’s superiority over ETM variant. Further, while ETM appears to do well on sampling, it’s empirical advantage over alternatives seems to be statistically insignificant from Table 2. While I am willing to believe the hypothesis that ETM is infact superior, further experimentation is clearly needed to support that. Lastly, the sampling experiment doesn’t seem to be actually testing sampling from a tilted distribution, but rather from the same distribution at a different temperature. This can be seen by substituting E_0=E_1/T_0. This presumably is much easier to do than if E_0 was an entirely different density, e.g. a Gaussian. Similarly, ITM is proven to offer better variance than WFM, however, this is not empirically demonstrated. More importantly, it is not clear from experiments that the theoretical result actually translates into any practical gain. Overall, in my view, the experiments just don’t support the theoretical results yet, and significant non-trivial work is needed to establish that.

- Lack of comparison to reweighted flow matching: reweigthing is an obvious and easy to implement alternative that doesn’t require much of the theoretical machinery, while offering a much simpler alternative. No evaluation is provided that reject this as a preferable alternative. Again, hinting towards an incomplete experimental evaluation.

Overall, while the theory is promising, I think the paper is severely lacking in empirical validation and needs significant amount of work to support the hypothesized advantages of theoretical results.

**Questions:**

Please see the weaknesses section for key issues.

---

> ### Author Response · Authors · 2025-11-27
>
> Thank you for your review. We address your concerns and questions below.
>
> ### ETM vs. ITM performance
>
> The reviewer is correct that ETM appears to outperform ITM in the LJ13 sampling experiments. This difference, however, is due to the use of an adaptive step size for ETM. Specifically, ETM dynamically adjusts its step size by computing the effective sample size (ESS) at every iteration. This procedure is computationally expensive but substantially improves stability and accuracy. By contrast, ITM was run with a fixed step size of 0.001.
>
> Since the original manuscript, we have incorporated a stopgrad in the ITM objective. This has led to improved stability and results for fine-tuning on Stable Diffusion using ITM. We have incorporated these into the revised manuscript. The values are as follows:
>
> Avg ImageReward : 0.4465 ± 0.0709
>
> Avg CLIP-Score  : 0.2794 ± 0.0036
>
> Avg HPS         : 0.2659 ± 0.0027
>
> Avg Aesthetic   : 6.0332 ± 0.0658
>
> Avg DreamSim Var: 0.3383 ± 0.0116
>
> Please note that there is a clear increase in ImageReward score for ITM over ETM while the other scores remain similar.
>
>
> ### Strength of Experimental Results
>
> In fine-tuning experiments, a common issue is "reward hacking" where optimizing solely for a single reward leads to degradation in other perceptual quality metrics. Our purpose in Table 2 is to show precisely that this does not occur: while we train *only* on ImageReward, other quality metrics remain stable and do not deteriorate. We will clarify this intent in the paper, as the current text can be misread as reporting minimal improvements rather than demonstrating robustness against reward hacking. We also now include ImageReward scores.
>
> As such, ITM and ETM achieve state-of-the-art results for diffusion-based sampling from the LJ potentials and on fine-tuning Stable Diffusion with ImageReward. We hope that together these demonstrate the practical gains of Tilt Matching.
>
>
>
> ### Temperature annealing path vs tilting
>
> The reviewer notes that the sampling experiment resembles annealing the temperature of the same distribution rather than tilting with a distinct reward. This is intentional: temperature annealing *is* a form of tilting a distribution. If the reviewer could help clarify what they mean, that would be helpful, because this paradigm easily fits into the tilted sampling we have devised. Temperature annealing is standard practice in the Boltzmann sampling literature and is used in many recent neural sampler approaches (e.g., PITA, PTSD). While this setup is easier than tilting between unrelated densities, it remains a widely-accepted and meaningful benchmark. This is a feature of our method, not a bug. Importantly, our method does not rely on multiple networks like PITA does, so if we choose to use importance weights, they are not biased. In PITA, each iteration of these weights introduces a new form of bias.
>
>
> ### Weighted Flow Matching
>
> We would like to clarify that we are not claiming that weighted flow matching (WFM) is a poor or incorrect algorithm. In fact, WFM is mathematically a special case of our framework, obtained by choosing the control variate $c=0$. Our theoretical discussion of WFM was included solely to motivate the broader family of ITM objectives and to highlight *why other instantiations can enjoy improved stability*. The higher variance of WFM implies that, in practice, it typically requires smaller learning rates to remain stable. For example, we found that WFM could diverge if the learning rate was not decreased within each annealing step from $a$ to $a+h$, rendering the resultant sampler useless. By contrast, ITM remains stable under larger step sizes, making it a more robust method. However, with a well-tuned learning rate schedule, both methods perform well. The higher variance objective simply means that training is less stable. We provide an example of the difference in ESS under this effect in Figure 5 in the appendix C.2.
>
>
> Please let us know if this addresses your concerns regarding experimental validation. We see that both ETM and ITM are robust on fine-tuning, surpassing the state of the art for fine-tuning flows/diffusions with a straightforward algorithm. We have updated the text accordingly and would appreciate any requisite adjustment in score. :)

---

### Author Response · Authors · 2025-11-27

We thank the reviewers for the detailed feedback and for highlighting both the strengths and the areas where clarification and additional results are needed. Below we address the main concerns and then respond individually.

**Motivation**

A key appeal of neural samplers is their potential to amortize the cost of learning samplers over large ensembles of target energy functions, so that a single trained model can be reused across many related systems. This is also the motivation behind several prior neural samplers, such as Adjoint Sampling. Our work takes a step toward a systematic methodology for this goal with the Tilt Matching framework. Accordingly, our Lennard–Jones experiments are set up to probe this regime and to illustrate that performance systematically improves as we allocate more compute, indicating that accuracy is primarily limited by compute rather than by the method itself.

**WFM vs ITM**

We want to clarify that weighted flow matching (WFM) is a special case of our framework, and that we are not advising against it as a training method. However, the higher variance of WFM implies that, in practice, it typically requires smaller learning rates to remain stable. By contrast, ITM remains stable under larger step sizes, making it a more robust method. We provide an example of the difference in ESS evolution with training in the appendix C.2.

**Table 2 ITM + ImageReward**

We have added the ImageReward scores to the table and included a new ITM run. In Table 2 of the updated manuscript, please note that both ETM and ITM outperform AM at same level of tilt in just a few hours of training on 4 A100s. Furthermore, the ClipScore and HPSv2 scores increase as well while we train *only* on ImageReward.

**Experimental Details**

We have included a cleaner discussion of the experimental details in the revised paper.

**Regularity of Transport Plan**

Our approach does *not* transport along the energy interpolation path. Instead, the drift $b_{t,a}$ always transports samples along the stochastic interpolant path, which is the fixed linear interpolation from the Gaussian prior $p_0$ to the current target $p_{1,a}$. This path is well known to have favorable regularity and does not exhibit mode-switching behavior.

Please let us know if there are any other questions or if you have any further suggestions, and we would be grateful if you could consider adjusting your score accordingly.

---

### Meta-Review · Area_Chair_73fV · 2025-12-28

**Summary:**

The reviewers generally agreed that the paper’s main idea, Tilt Matching, is theoretically interesting and novel. However, they had several common worries that led to a lean toward rejection:

Weak Evidence for Practical Gain: Reviewers felt the experiments didn't fully prove that the new "ITM" method is actually better in practice than simpler existing methods.

Missing Baselines: The paper compared Tilt Matching mostly to other "neural samplers" but ignored classic, high-performing sampling methods used in science.

Simple Test Cases Missing: Reviewers strongly requested tests on "Gaussian Mixture Models" (a standard way to see if a sampler can find multiple separate "pockets" of data), which were not included.

Computational Cost: It wasn't clear if the method is actually faster or more efficient when you factor in the time needed for its iterative training steps.

**Reviewer Concerns:**

Addressed by the Rebuttal:

Comparison to ASBS: The authors updated the paper to include a comparison with a strong baseline called ASBS, showing their method performed better on some metrics.

Missing Scores: The authors added "ImageReward" scores to their fine-tuning table and provided a missing mathematical derivation (Equation 26).

Procedural Clarity: They cleared up confusion about a "refresh" step and the "mode-switching" issue by providing better explanations and removing confusing references.

Discretization Study: They added a new figure (Figure 7) showing how their method’s accuracy improves as you give it more computing time.

Still Outstanding:

Gaussian Mixture Model (GMM) Tests: The authors argued that their current tests were enough, but reviewers QPg2 and PMF8 flatly disagreed, stating these simple tests are necessary to judge a sampler's reliability.

Classical Sampler Baselines: Reviewer PMF8 insisted that without comparing to established non-neural methods (like Parallel Tempering), it’s impossible to tell if this new method is actually useful for real-world science.

Stability Proof: Some reviewers remained skeptical that the theoretical "low variance" actually makes training easier in a meaningful way.

**Reviewer Scores:**

For Reviewer dbLF, the score would likely have maintained the same at a 2 (Reject). Although they acknowledged the theoretical interestingness of the "Covariance ODE," they remained highly skeptical of the practical gains. They specifically noted that the empirical evidence didn't clearly demonstrate superiority over simpler baselines, and the updated results in the rebuttal weren't enough to overcome their concerns about the limited scale of the sampling experiments.

Reviewer QPg2 would also likely have maintained the same score of 4 (Below Threshold). This reviewer took a very firm stance, explicitly stating in the late-stage discussion that they were unwilling to raise their score without systematic experiments on Gaussian Mixture Models. While the authors provided a new scaling plot (Figure 7), the reviewer maintained that without testing on multimodal targets to prove the method can find separate "pockets" of data, the evaluation remained incomplete.

In contrast, Reviewer ojYB is the most likely to have increased their score. This reviewer provided a very specific list of requests, including the addition of the ASBS baseline, missing math derivations, and ImageReward scores. The authors addressed every one of these points directly in the rebuttal and updated the manuscript accordingly, which typically leads to a more favorable final assessment.

Finally, Reviewer PMF8 would almost certainly have maintained the same score of 2 (Reject). This reviewer was the most critical of the sampling evaluation, emphasizing that the paper ignored established non-neural sampling techniques like Parallel Tempering. Like Reviewer QPg2, they were not satisfied with the authors' refusal to run Gaussian Mixture Model tests, concluding the discussion by explicitly stating they continued to recommend rejection.

---

### Decision · Program_Chairs · 2026-01-26

Reject